# PROGRESSIVE DISTILLATION FOR FAST SAMPLING OF DIFFUSION MODELS

**Tim Salimans & Jonathan Ho**
Google Research, Brain team
{salimans,jonathanho}@google.com

## ABSTRACT

Diffusion models have recently shown great promise for generative modeling, outperforming GANs on perceptual quality and autoregressive models at density estimation. A remaining downside is their slow sampling time: generating high quality samples takes many hundreds or thousands of model evaluations. Here we make two contributions to help eliminate this downside: First, we present new parameterizations of diffusion models that provide increased stability when using few sampling steps. Second, we present a method to distill a trained deterministic diffusion sampler, using many steps, into a new diffusion model that takes half as many sampling steps. We then keep progressively applying this distillation procedure to our model, halving the number of required sampling steps each time. On standard image generation benchmarks like CIFAR-10, ImageNet, and LSUN, we start out with state-of-the-art samplers taking as many as 8192 steps, and are able to distill down to models taking as few as 4 steps without losing much perceptual quality; achieving, for example, a FID of 3.0 on CIFAR-10 in 4 steps. Finally, we show that the full progressive distillation procedure does not take more time than it takes to train the original model, thus representing an efficient solution for generative modeling using diffusion at both train and test time.

## 1 INTRODUCTION

Diffusion models (Sohl-Dickstein et al., 2015; Song & Ermon, 2019; Ho et al., 2020) are an emerging class of generative models that has recently delivered impressive results on many standard generative modeling benchmarks. These models have achieved ImageNet generation results outperforming BigGAN-deep and VQ-VAE-2 in terms of FID score and classification accuracy score (Ho et al., 2021; Dhariwal & Nichol, 2021), and they have achieved likelihoods outperforming autoregressive image models (Kingma et al., 2021; Song et al., 2021b). They have also succeeded in image super-resolution (Saharia et al., 2021; Li et al., 2021) and image inpainting (Song et al., 2021c), and there have been promising results in shape generation (Cai et al., 2020), graph generation (Niu et al., 2020), and text generation (Hoogeboom et al., 2021; Austin et al., 2021).

A major barrier remains to practical adoption of diffusion models: sampling speed. While sampling can be accomplished in relatively few steps in strongly conditioned settings, such as text-to-speech (Chen et al., 2021) and image super-resolution (Saharia et al., 2021), or when guiding the sampler using an auxiliary classifier (Dhariwal & Nichol, 2021), the situation is substantially different in settings in which there is less conditioning information available. Examples of such settings are unconditional and standard class-conditional image generation, which currently require hundreds or thousands of steps using network evaluations that are not amenable to the caching optimizations of other types of generative models (Ramachandran et al., 2017).

In this paper, we reduce the sampling time of diffusion models by orders of magnitude in unconditional and class-conditional image generation, which represent the setting in which diffusion models have been slowest in previous work. We present a procedure to distill the behavior of a $N$-step DDIM sampler (Song et al., 2021a) for a pretrained diffusion model into a new model with $N/2$ steps, with little degradation in sample quality. In what we call *progressive distillation*, we repeat this distillation procedure to produce models that generate in as few as 4 steps, still maintaining sample quality competitive with state-of-the-art models using thousands of steps.

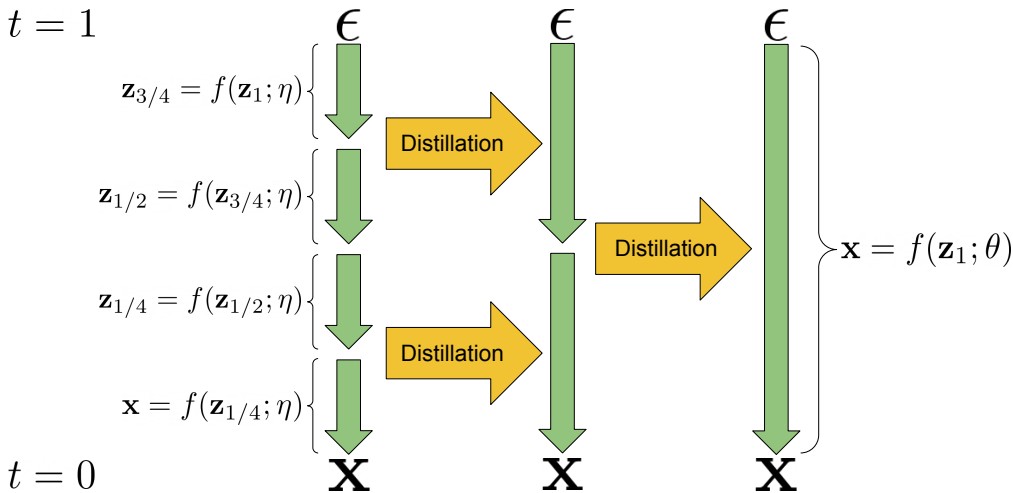

Figure 1: A visualization of two iterations of our proposed *progressive distillation* algorithm. A sampler $f(\mathbf{z}; \eta)$, mapping random noise $\epsilon$ to samples $\mathbf{x}$ in 4 deterministic steps, is distilled into a new sampler $f(\mathbf{z}; \theta)$ taking only a single step. The original sampler is derived by approximately integrating the *probability flow ODE* for a learned diffusion model, and distillation can thus be understood as learning to integrate in fewer steps, or *amortizing* this integration into the new sampler.

## 2 BACKGROUND ON DIFFUSION MODELS

We consider diffusion models (Sohl-Dickstein et al., 2015; Song & Ermon, 2019; Ho et al., 2020) specified in continuous time (Tzen & Raginsky, 2019a; Song et al., 2021c; Chen et al., 2021; Kingma et al., 2021). We use $\mathbf{x} \sim p(\mathbf{x})$ to denote training data. A diffusion model has latent variables $\mathbf{z} = \{\mathbf{z}_t \,|\, t \in [0, 1]\}$ and is specified by a noise schedule comprising differentiable functions $\alpha_t, \sigma_t$ such that $\lambda_t = \log[\alpha_t^2/\sigma_t^2]$, the log signal-to-noise-ratio, decreases monotonically with $t$.

These ingredients define the forward process $q(\mathbf{z}|\mathbf{x})$, a Gaussian process satisfying the following Markovian structure:

$$q(\mathbf{z}_t|\mathbf{x}) = \mathcal{N}(\mathbf{z}_t; \alpha_t\mathbf{x}, \sigma_t^2\mathbf{I}), \quad q(\mathbf{z}_t|\mathbf{z}_s) = \mathcal{N}(\mathbf{z}_t; (\alpha_t/\alpha_s)\mathbf{z}_s, \sigma_{t|s}^2\mathbf{I}) \tag{1}$$

where $0 \le s < t \le 1$ and $\sigma_{t|s}^2 = (1 - e^{\lambda_t - \lambda_s})\sigma_t^2$.

The role of function approximation in the diffusion model is to denoise $\mathbf{z}_t \sim q(\mathbf{z}_t|\mathbf{x})$ into an estimate $\hat{\mathbf{x}}_\theta(\mathbf{z}_t) \approx \mathbf{x}$ (the function approximator also receives $\lambda_t$ as an input, but we omit this to keep our notation clean). We train this denoising model $\hat{\mathbf{x}}_\theta$ using a weighted mean squared error loss

$$\mathbb{E}_{\epsilon, t}\big[w(\lambda_t)\|\hat{\mathbf{x}}_\theta(\mathbf{z}_t) - \mathbf{x}\|_2^2\big] \tag{2}$$

over uniformly sampled times $t \in [0, 1]$. This loss can be justified as a weighted variational lower bound on the data log likelihood under the diffusion model (Kingma et al., 2021) or as a form of denoising score matching (Vincent, 2011; Song & Ermon, 2019). We will discuss particular choices of weighting function $w(\lambda_t)$ later on.

Sampling from a trained model can be performed in several ways. The most straightforward way is discrete time ancestral sampling (Ho et al., 2020). To define this sampler, first note that the forward process can be described in reverse as $q(\mathbf{z}_s|\mathbf{z}_t, \mathbf{x}) = \mathcal{N}(\mathbf{z}_s; \tilde{\boldsymbol{\mu}}_{s|t}(\mathbf{z}_t, \mathbf{x}), \tilde{\sigma}_{s|t}^2\mathbf{I})$ (noting $s < t$), where

$$\tilde{\boldsymbol{\mu}}_{s|t}(\mathbf{z}_t, \mathbf{x}) = e^{\lambda_t - \lambda_s}(\alpha_s/\alpha_t)\mathbf{z}_t + (1 - e^{\lambda_t - \lambda_s})\alpha_s\mathbf{x}, \quad \tilde{\sigma}_{s|t}^2 = (1 - e^{\lambda_t - \lambda_s})\sigma_s^2 \tag{3}$$

We use this reversed description of the forward process to define the ancestral sampler. Starting at $\mathbf{z}_1 \sim \mathcal{N}(\mathbf{0}, \mathbf{I})$, the ancestral sampler follows the rule

$$\mathbf{z}_s = \tilde{\boldsymbol{\mu}}_{s|t}(\mathbf{z}_t, \hat{\mathbf{x}}_\theta(\mathbf{z}_t)) + \sqrt{(\tilde{\sigma}_{s|t}^2)^{1-\gamma}(\sigma_{t|s}^2)^\gamma}\,\epsilon \tag{4}$$

$$= e^{\lambda_t - \lambda_s}(\alpha_s/\alpha_t)\mathbf{z}_t + (1 - e^{\lambda_t - \lambda_s})\alpha_s\hat{\mathbf{x}}_\theta(\mathbf{z}_t) + \sqrt{(\tilde{\sigma}_{s|t}^2)^{1-\gamma}(\sigma_{t|s}^2)^\gamma}\,\epsilon, \tag{5}$$

where $\epsilon$ is standard Gaussian noise, and $\gamma$ is a hyperparameter that controls how much noise is added during sampling, following Nichol & Dhariwal (2021).

Alternatively, Song et al. (2021c) show that our denoising model $\hat{\mathbf{x}}_\theta(\mathbf{z}_t)$ can be used to deterministically map noise $\mathbf{z}_1 \sim \mathcal{N}(\mathbf{0}, \mathbf{I})$ to samples $\mathbf{x}$ by numerically solving the *probability flow ODE*:

$$d\mathbf{z}_t = [f(\mathbf{z}_t, t) - \frac{1}{2}g^2(t)\nabla_z \log \hat{p}_\theta(\mathbf{z}_t)]dt, \tag{6}$$

where $\nabla_z \log \hat{p}_\theta(\mathbf{z}_t) = \frac{\alpha_t \hat{\mathbf{x}}_\theta(\mathbf{z}_t) - \mathbf{z}_t}{\sigma_t^2}$. Following Kingma et al. (2021), we have $f(\mathbf{z}_t, t) = \frac{d \log \alpha_t}{dt}\mathbf{z}_t$ and $g^2(t) = \frac{d\sigma_t^2}{dt} - 2\frac{d \log \alpha_t}{dt}\sigma_t^2$. Since $\hat{\mathbf{x}}_\theta(\mathbf{z}_t)$ is parameterized by a neural network, this equation is a special case of a *neural ODE* (Chen et al., 2018), also called a *continuous normalizing flow* (Grathwohl et al., 2018).

Solving the ODE in Equation 6 numerically can be done with standard methods like the Euler rule or the Runge-Kutta method. The DDIM sampler proposed by Song et al. (2021a) can also be understood as an integration rule for this ODE, as we show in Appendix B, even though it was originally proposed with a different motivation. The update rule specified by DDIM is

$$\mathbf{z}_s = \alpha_s \hat{\mathbf{x}}_\theta(\mathbf{z}_t) + \sigma_s \frac{\mathbf{z}_t - \alpha_t \hat{\mathbf{x}}_\theta(\mathbf{z}_t)}{\sigma_t} \tag{7}$$

$$= e^{(\lambda_t - \lambda_s)/2}(\alpha_s/\alpha_t)\mathbf{z}_t + (1 - e^{(\lambda_t - \lambda_s)/2})\alpha_s \hat{\mathbf{x}}_\theta(\mathbf{z}_t), \tag{8}$$

and in practice this rule performs better than the aforementioned standard ODE integration rules in our case, as we show in Appendix C.

If $\hat{\mathbf{x}}_\theta(\mathbf{z}_t)$ satisfies mild smoothness conditions, the error introduced by numerical integration of the probability flow ODE is guaranteed to vanish as the number of integration steps grows infinitely large, i.e. $N \to \infty$. This leads to a trade-off in practice between the accuracy of the numerical integration, and hence the quality of the produced samples from our model, and the time needed to produce these samples. So far, most models in the literature have needed hundreds or thousands of integration steps to produce their highest quality samples, which is prohibitive for many practical applications of generative modeling. Here, we therefore propose a method to distill these accurate, but slow, ODE integrators into much faster models that are still very accurate. This idea is visualized in Figure 1, and described in detail in the next section.

## 3 PROGRESSIVE DISTILLATION

To make diffusion models more efficient at sampling time, we propose *progressive distillation*: an algorithm that iteratively halves the number of required sampling steps by distilling a slow teacher diffusion model into a faster student model. Our implementation of progressive distillation stays very close to the implementation for training the original diffusion model, as described by e.g. Ho et al. (2020). Algorithm 1 and Algorithm 2 present diffusion model training and progressive distillation side-by-side, with the relative changes in progressive distillation highlighted in green.

We start the progressive distillation procedure with a teacher diffusion model that is obtained by training in the standard way. At every iteration of progressive distillation, we then initialize the student model with a copy of the teacher, using both the same parameters and same model definition. Like in standard training, we then sample data from the training set and add noise to it, before forming the training loss by applying the student denoising model to this noisy data $\mathbf{z}_t$. The main difference in progressive distillation is in how we set the target for the denoising model: instead of the original data $\mathbf{x}$, we have the student model denoise towards a target $\tilde{\mathbf{x}}$ that makes a single student DDIM step match 2 teacher DDIM steps. We calculate this target value by running 2 DDIM sampling steps using the teacher, starting from $\mathbf{z}_t$ and ending at $\mathbf{z}_{t-1/N}$, with $N$ being the number of student sampling steps. By inverting a single step of DDIM, we then calculate the value the student model would need to predict in order to move from $\mathbf{z}_t$ to $\mathbf{z}_{t-1/N}$ in a single step, as we show in detail in Appendix G. The resulting target value $\tilde{\mathbf{x}}(\mathbf{z}_t)$ is fully determined given the teacher model and starting point $\mathbf{z}_t$, which allows the student model to make a sharp prediction when evaluated at $\mathbf{z}_t$. In contrast, the original data point $\mathbf{x}$ is not fully determined given $\mathbf{z}_t$, since multiple different data points $\mathbf{x}$ can produce the same noisy data $\mathbf{z}_t$: this means that the original denoising model is

predicting a weighted average of possible $\mathbf{x}$ values, which produces a blurry prediction. By making sharper predictions, the student model can make faster progress during sampling.

After running distillation to learn a student model taking $N$ sampling steps, we can repeat the procedure with $N/2$ steps: The student model then becomes the new teacher, and a new student model is initialized by making a copy of this model.

Unlike our procedure for training the original model, we always run progressive distillation in discrete time: we sample this discrete time such that the highest time index corresponds to a signal-to-noise ratio of zero, i.e. $\alpha_1 = 0$, which exactly matches the distribution of input noise $\mathbf{z}_1 \sim \mathcal{N}(\mathbf{0}, \mathbf{I})$ that is used at test time. We found this to work slightly better than starting from a non-zero signal-to-noise ratio as used by e.g. Ho et al. (2020), both for training the original model as well as when performing progressive distillation.

---

**Algorithm 1** Standard diffusion training

**Require:** Model $\hat{\mathbf{x}}_\theta(\mathbf{z}_t)$ to be trained
**Require:** Data set $\mathcal{D}$
**Require:** Loss weight function $w()$

**while** not converged **do**
    $\mathbf{x} \sim \mathcal{D}$     ▷ Sample data
    $t \sim U[0, 1]$     ▷ Sample time
    $\epsilon \sim N(0, I)$     ▷ Sample noise
    $\mathbf{z}_t = \alpha_t \mathbf{x} + \sigma_t \epsilon$     ▷ Add noise to data

    $\tilde{\mathbf{x}} = \mathbf{x}$     ▷ Clean data is target for $\hat{\mathbf{x}}$
    $\lambda_t = \log[\alpha_t^2/\sigma_t^2]$     ▷ log-SNR
    $L_\theta = w(\lambda_t)\|\tilde{\mathbf{x}} - \hat{\mathbf{x}}_\theta(\mathbf{z}_t)\|_2^2$     ▷ Loss
    $\theta \leftarrow \theta - \gamma \nabla_\theta L_\theta$     ▷ Optimization
**end while**

---

**Algorithm 2** Progressive distillation

**Require:** Trained teacher model $\hat{\mathbf{x}}_\eta(\mathbf{z}_t)$
**Require:** Data set $\mathcal{D}$
**Require:** Loss weight function $w()$
**Require:** Student sampling steps $N$
  **for** $K$ iterations **do**
    $\theta \leftarrow \eta$     ▷ Init student from teacher
    **while** not converged **do**
      $\mathbf{x} \sim \mathcal{D}$
      $t = i/N, \ \ i \sim Cat[1, 2, \ldots, N]$
      $\epsilon \sim N(0, I)$
      $\mathbf{z}_t = \alpha_t \mathbf{x} + \sigma_t \epsilon$
      `# 2 steps of DDIM with teacher`
      $t' = t - 0.5/N, \ \ t'' = t - 1/N$
      $\mathbf{z}_{t'} = \alpha_{t'}\hat{\mathbf{x}}_\eta(\mathbf{z}_t) + \frac{\sigma_{t'}}{\sigma_t}(\mathbf{z}_t - \alpha_t\hat{\mathbf{x}}_\eta(\mathbf{z}_t))$
      $\mathbf{z}_{t''} = \alpha_{t''}\hat{\mathbf{x}}_\eta(\mathbf{z}_{t'}) + \frac{\sigma_{t''}}{\sigma_{t'}}(\mathbf{z}_{t'} - \alpha_{t'}\hat{\mathbf{x}}_\eta(\mathbf{z}_{t'}))$
      $\tilde{\mathbf{x}} = \frac{\mathbf{z}_{t''} - (\sigma_{t''}/\sigma_t)\mathbf{z}_t}{\alpha_{t''} - (\sigma_{t''}/\sigma_t)\alpha_t}$     ▷ Teacher $\hat{\mathbf{x}}$ target
      $\lambda_t = \log[\alpha_t^2/\sigma_t^2]$
      $L_\theta = w(\lambda_t)\|\tilde{\mathbf{x}} - \hat{\mathbf{x}}_\theta(\mathbf{z}_t)\|_2^2$
      $\theta \leftarrow \theta - \gamma \nabla_\theta L_\theta$
    **end while**
    $\eta \leftarrow \theta$     ▷ Student becomes next teacher
    $N \leftarrow N/2$  ▷ Halve number of sampling steps
  **end for**

---

## 4 DIFFUSION MODEL PARAMETERIZATION AND TRAINING LOSS

In this section, we discuss how to parameterize the denoising model $\hat{\mathbf{x}}_\theta$, and how to specify the reconstruction loss weight $w(\lambda_t)$. We assume a standard *variance-preserving* diffusion process for which $\sigma_t^2 = 1 - \alpha_t^2$. This is without loss of generality, as shown by (Kingma et al., 2021, appendix G): different specifications of the diffusion process, such as the *variance-exploding* specification, can be considered equivalent to this specification, up to rescaling of the noisy latents $\mathbf{z}_t$. We use a cosine schedule $\alpha_t = \cos(0.5\pi t)$, similar to that introduced by Nichol & Dhariwal (2021).

Ho et al. (2020) and much of the following work choose to parameterize the denoising model through directly predicting $\epsilon$ with a neural network $\hat{\epsilon}_\theta(\mathbf{z}_t)$, which implicitly sets $\hat{\mathbf{x}}_\theta(\mathbf{z}_t) = \frac{1}{\alpha_t}(\mathbf{z}_t - \sigma_t\hat{\epsilon}_\theta(\mathbf{z}_t))$. In this case, the training loss is also usually defined as mean squared error in the $\epsilon$-space:

$$L_\theta = \|\epsilon - \hat{\epsilon}_\theta(\mathbf{z}_t)\|_2^2 = \left\|\frac{1}{\sigma_t}(\mathbf{z}_t - \alpha_t\mathbf{x}) - \frac{1}{\sigma_t}(\mathbf{z}_t - \alpha_t\hat{\mathbf{x}}_\theta(\mathbf{z}_t))\right\|_2^2 = \frac{\alpha_t^2}{\sigma_t^2}\|\mathbf{x} - \hat{\mathbf{x}}_\theta(\mathbf{z}_t)\|_2^2, \quad (9)$$

which can thus equivalently be seen as a weighted reconstruction loss in $\mathbf{x}$-space, where the weighting function is given by $w(\lambda_t) = \exp(\lambda_t)$, for log signal-to-noise ratio $\lambda_t = \log[\alpha_t^2/\sigma_t^2]$.

Although this standard specification works well for training the original model, it is not well suited for distillation: when training the original diffusion model, and at the start of progressive distillation, the model is evaluated at a wide range of signal-to-noise ratios $\alpha_t^2/\sigma_t^2$, but as distillation progresses we increasingly evaluate at lower and lower signal-to-noise ratios. As the signal-to-noise ratio goes to zero, the effect of small changes in the neural network output $\hat{\epsilon}_\theta(\mathbf{z}_t)$ on the implied prediction in $\mathbf{x}$-space is increasingly amplified, since $\hat{\mathbf{x}}_\theta(\mathbf{z}_t) = \frac{1}{\alpha_t}(\mathbf{z}_t - \sigma_t\hat{\epsilon}_\theta(\mathbf{z}_t))$ divides by $\alpha_t \rightarrow 0$. This is not much of a problem when taking many steps, since the effect of early missteps is limited by clipping of the $\mathbf{z}_t$ iterates, and later updates can correct any mistakes, but it becomes increasingly important as we decrease the number of sampling steps. Eventually, if we distill all the way down to a single sampling step, the input to the model is only pure noise $\epsilon$, which corresponds to a signal-to-noise ratio of zero, i.e. $\alpha_t = 0, \sigma_t = 1$. At this extreme, the link between $\epsilon$-prediction and $\mathbf{x}$-prediction breaks down completely: observed data $\mathbf{z}_t = \epsilon$ is no longer informative of $\mathbf{x}$ and predictions $\hat{\epsilon}_\theta(\mathbf{z}_t)$ no longer implicitly predict $\mathbf{x}$. Examining our reconstruction loss (equation 9), we see that the weighting function $w(\lambda_t)$ gives zero weight to the reconstruction loss at this signal-to-noise ratio.

For distillation to work, we thus need to parameterize the diffusion model in a way for which the implied prediction $\hat{\mathbf{x}}_\theta(\mathbf{z}_t)$ remains stable as $\lambda_t = \log[\alpha_t^2/\sigma_t^2]$ varies. We tried the following options, and found all to work well with progressive distillation:

- Predicting $\mathbf{x}$ directly.
- Predicting both $\mathbf{x}$ and $\epsilon$, via separate output channels $\{\tilde{\mathbf{x}}_\theta(\mathbf{z}_t), \tilde{\epsilon}_\theta(\mathbf{z}_t)\}$ of the neural network, and then merging the predictions via $\hat{\mathbf{x}} = \sigma_t^2\tilde{\mathbf{x}}_\theta(\mathbf{z}_t) + \alpha_t(\mathbf{z}_t - \sigma_t\tilde{\epsilon}_\theta(\mathbf{z}_t))$, thus smoothly interpolating between predicting $\mathbf{x}$ directly and predicting via $\epsilon$.
- Predicting $\mathbf{v} \equiv \alpha_t\epsilon - \sigma_t\mathbf{x}$, which gives $\hat{\mathbf{x}} = \alpha_t\mathbf{z}_t - \sigma_t\hat{\mathbf{v}}_\theta(\mathbf{z}_t)$, as we show in Appendix D.

In Section 5.1 we test all three parameterizations on training an original diffusion model (no distillation), and find them to work well there also.

In addition to determining an appropriate parameterization, we also need to decide on a reconstruction loss weighting $w(\lambda_t)$. The setup of Ho et al. (2020) weights the reconstruction loss by the signal-to-noise ratio, implicitly gives a weight of zero to data with zero SNR, and is therefore not a suitable choice for distillation. We consider two alternative training loss weightings:

- $L_\theta = \max(\|\mathbf{x} - \hat{\mathbf{x}}_t\|_2^2, \|\epsilon - \hat{\epsilon}_t\|_2^2) = \max(\frac{\alpha_t^2}{\sigma_t^2}, 1)\|\mathbf{x} - \hat{\mathbf{x}}_t\|_2^2$; 'truncated SNR' weighting.
- $L_\theta = \|\mathbf{v}_t - \hat{\mathbf{v}}_t\|_2^2 = (1 + \frac{\alpha_t^2}{\sigma_t^2})\|\mathbf{x} - \hat{\mathbf{x}}_t\|_2^2$; 'SNR+1' weighting.

We examine both choices in our ablation study in Section 5.1, and find both to be good choices for training diffusion models. In practice, the choice of loss weighting also has to take into account how $\alpha_t, \sigma_t$ are sampled during training, as this sampling distribution strongly determines the weight the expected loss gives to each signal-to-noise ratio. Our results are for a cosine schedule $\alpha_t = \cos(0.5\pi t)$, where time is sampled uniformly from $[0, 1]$. In Figure 2 we visualize the resulting loss weightings, both including and excluding the effect of the cosine schedule.

## 5 EXPERIMENTS

In this section we empirically validate the progressive distillation algorithm proposed in Section 3, as well as the parameterizations and loss weightings considered in Section 4. We consider various image generation benchmarks, with resolution varying from $32 \times 32$ to $128 \times 128$. All experiments use the cosine schedule $\alpha_t = \cos(0.5\pi t)$, and all models use a U-Net architecture similar to that introduced by Ho et al. (2020), but with BigGAN-style up- and downsampling (Brock et al., 2019), as used in the diffusion modeling setting by Nichol & Dhariwal (2021); Song et al. (2021c). Our training setup closely matches the open source code by Ho et al. (2020). Exact details are given in Appendix E.

### 5.1 MODEL PARAMETERIZATION AND TRAINING LOSS

As explained in Section 4, the standard method of having our model predict $\epsilon$, and minimizing mean squared error in the $\epsilon$-space (Ho et al., 2020), is not appropriate for use with progressive distillation.

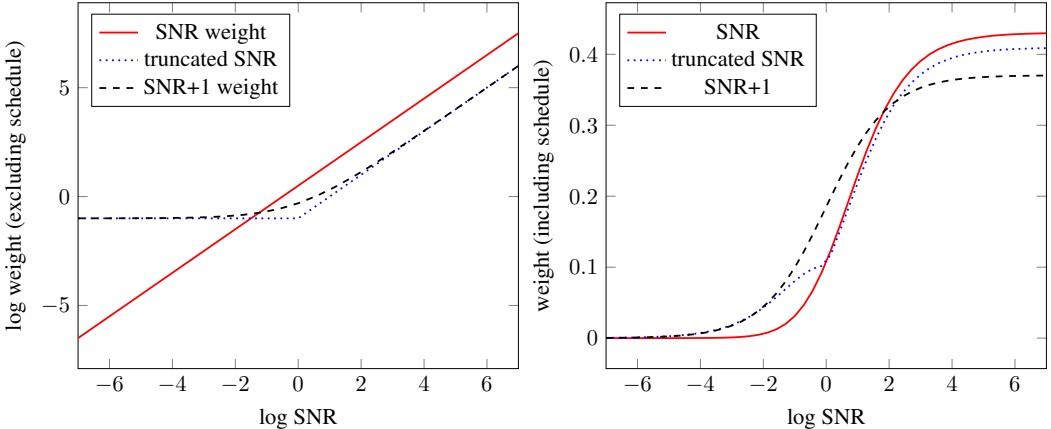

Figure 2: **Left:** Log weight assigned to reconstruction loss $\|\mathbf{x} - \hat{\mathbf{x}}_\lambda\|_2^2$ as a function of the log-SNR $\lambda = \log[\alpha^2/\sigma^2]$, for each of our considered training loss weightings, excluding the influence of the $\alpha_t, \sigma_t$ schedule. **Right:** Weights assigned to the reconstruction loss including the effect of the cosine schedule $\alpha_t = \cos(0.5\pi t)$, with $t \sim U[0, 1]$. The weights are only defined up to a constant, and we have adjusted these constants to fit this graph.

| Network Output | Loss Weighting | Stochastic sampler | DDIM sampler |
|---|---|---|---|
| $(\mathbf{x}, \epsilon)$ combined | SNR | 2.54/9.88 | 2.78/9.56 |
| | Truncated SNR | 2.47/9.85 | 2.76/9.49 |
| | SNR+1 | 2.52/9.79 | 2.87/9.45 |
| $\mathbf{x}$ | SNR | 2.65/9.80 | 2.75/9.56 |
| | Truncated SNR | 2.53/**9.92** | **2.51/9.58** |
| | SNR+1 | 2.56/9.84 | 2.65/9.52 |
| $\epsilon$ | SNR | 2.59/9.84 | 2.91/9.52 |
| | Truncated SNR | N/A | N/A |
| | SNR+1 | 2.56/9.77 | 3.27/9.41 |
| $\mathbf{v}$ | SNR | 2.65/9.86 | 3.05/9.56 |
| | Truncated SNR | **2.45**/9.80 | 2.75/9.52 |
| | SNR+1 | 2.49/9.77 | 2.87/9.43 |

Table 1: Generated sample quality as measured by FID and Inception Score (FID/IS) on unconditional CIFAR-10, training the original model (no distillation), and comparing different parameterizations and loss weightings discussed in Section 4. All reported results are averages over 3 random seeds of the best metrics obtained over 2 million training steps; nevertheless we find results are still $\pm 0.1$ due to the noise inherent in training our models. Taking the neural network output to represent a prediction of $\epsilon$ in combination with the Truncated SNR loss weighting leads to divergence.

We therefore proposed various alternative parameterizations of the denoising diffusion model that are stable under the progressive distillation procedure, as well as various weighting functions for the reconstruction error in $\mathbf{x}$-space. Here, we perform a complete ablation experiment of all parameterizations and loss weightings considered in Section 4. For computational efficiency, and for comparisons to established methods in the literature, we use unconditional CIFAR-10 as the benchmark. We measure performance of undistilled models trained from scratch, to avoid introducing too many factors of variation into our analysis.

Table 1 lists the results of the ablation study. Overall results are fairly close across different parameterizations and loss weights. All proposed stable model specifications achieve excellent performance, with the exception of the combination of outputting $\epsilon$ with the neural network and weighting the loss with the truncated SNR, which we find to be unstable. Both predicting $\mathbf{x}$ directly, as well as predicting $\mathbf{v}$, or the combination $(\epsilon, \mathbf{x})$, could thus be recommended for specification of diffusion

models. Here, predicting **v** is the most stable option, as it has the unique property of making DDIM step-sizes independent of the SNR (see Appendix D), but predicting **x** gives slightly better empirical results in this ablation study.

## 5.2 PROGRESSIVE DISTILLATION

We evaluate our proposed progressive distillation algorithm on 4 data sets: CIFAR-10, $64 \times 64$ downsampled ImageNet, $128 \times 128$ LSUN bedrooms, and $128 \times 128$ LSUN Church-Outdoor. For each data set we start by training a baseline model, after which we start the progressive distillation procedure. For CIFAR-10 we start progressive distillation from a teacher model taking 8192 steps. For the bigger data sets we start at 1024 steps. At every iteration of distillation we train for 50 thousand parameter updates, except for the distillation to 2 and 1 sampling steps, for which we use 100 thousand updates. We report FID results obtained after each iteration of the algorithm. Using these settings, the computational cost of progressive distillation to 4 sampling steps is comparable or less than for training the original model. In Appendix I we show that this computational cost can be reduce much further still, at a small cost in performance.

In Figure 4 we plot the resulting FID scores (Heusel et al., 2017) obtained for each number of sampling steps. We compare against the undistilled DDIM sampler, as well as to a highly optimized stochastic baseline sampler. For all four data sets, progressive distillation produces near optimal results up to 4 or 8 sampling steps. At 2 or 1 sampling steps, the sample quality degrades relatively more quickly. In contrast, the quality of the DDIM and stochastic samplers degrades very sharply after reducing the number of sampling steps below 128. Overall, we conclude that progressive distillation is thus an attractive solution for computational budgets that allow less than or equal to 128 sampling steps. Although our distillation procedure is designed for use with the DDIM sampler, the resulting distilled models can in principle also be used with stochastic sampling: we investigate this in Appendix F, and find that it achieves performance that falls in between the distilled DDIM sampler and the undistilled stochastic sampler.

Table 2 shows some of our results on CIFAR-10, and compares against other fast sampling methods in the literature: Our method compares favorably and attains higher sampling quality in fewer steps than most of the alternative methods. Figure 3 shows some random samples from our model obtained at different phases of the distillation process. Additional samples are provided in Appendix H.

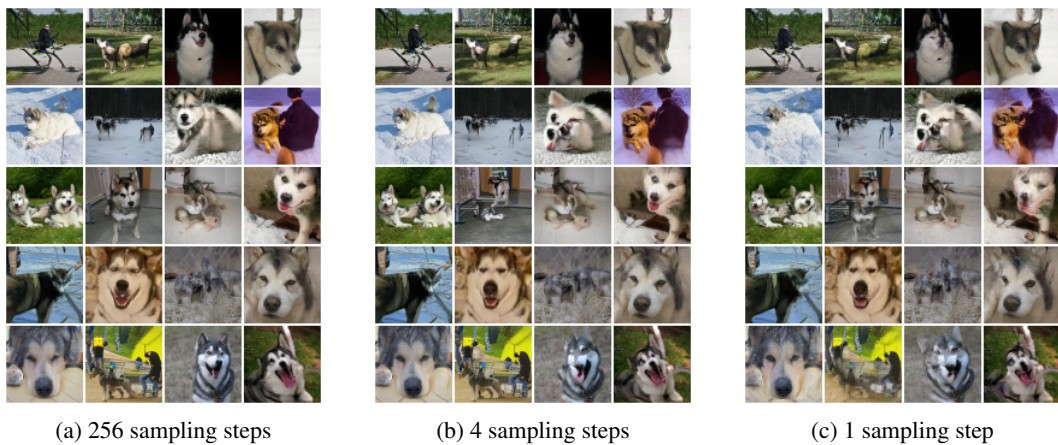

    (a) 256 sampling steps          (b) 4 sampling steps          (c) 1 sampling step

Figure 3: Random samples from our distilled $64 \times 64$ ImageNet models, conditioned on the 'mala-mute' class, for fixed random seed and for varying number of sampling steps. The mapping from input noise to output image is well preserved as the number of sampling steps is reduced.

## 6 RELATED WORK ON FAST SAMPLING

Our proposed method is closest to the work of Luhman & Luhman (2021), who perform distillation of DDIM teacher models into one-step student models. A possible downside of their method is

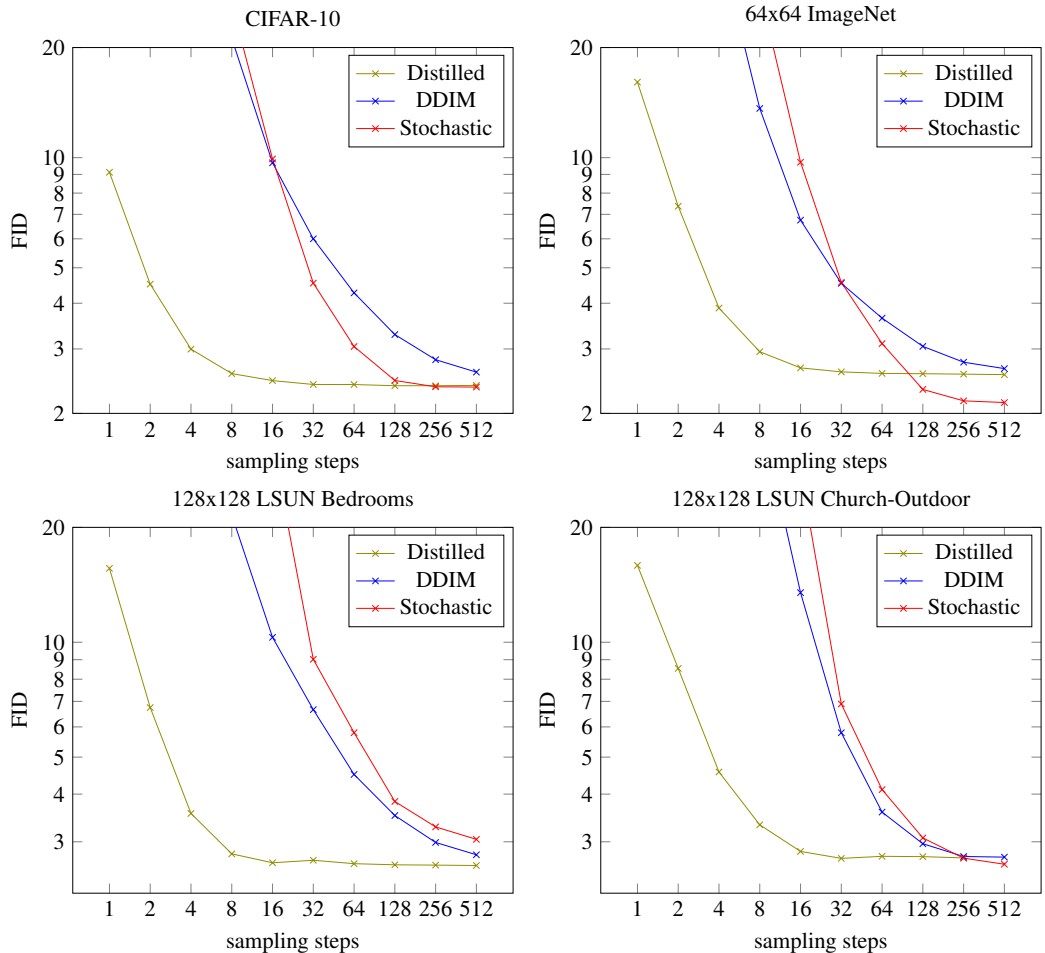

Figure 4: Sample quality results as measured by FID for our distilled model on unconditional CIFAR-10, class-conditional 64x64 ImageNet, 128x128 LSUN bedrooms, and 128x128 LSUN church-outdoor. We compare against the DDIM sampler and against an optimized stochastic sampler, each evaluated using the same models that were used to initialize the progressive distillation procedure. For CIFAR-10 we report an average over 4 random seeds. For the other data sets we only use a single run because of their computational demand. For the stochastic sampler we set the variance as a log-scale interpolation between an upper and lower bound on the variance, following Nichol & Dhariwal (2021), but we use a single interpolation coefficient rather than a learned coefficient. We then tune this interpolation coefficient separately for each number of sampling steps and report only the best result for that number of steps: this way we obtained better results than with the learned interpolation.

that it requires constructing a large data set by running the original model at its full number of sampling steps: their cost of distillation thus scales linearly with this number of steps, which can be prohibitive. In contrast, our method never needs to run the original model at the full number of sampling steps: at every iteration of progressive distillation, the number of model evaluations is independent of the number of teacher sampling steps, allowing our method to scale up to large numbers of teacher steps at a logarithmic cost in total distillation time.

DDIM (Song et al., 2021a) was originally shown to be effective for few-step sampling, as was the probability flow sampler (Song et al., 2021c). Jolicoeur-Martineau et al. (2021) study fast SDE integrators for reverse diffusion processes, and Tzen & Raginsky (2019b) study unbiased samplers which may be useful for fast, high quality sampling as well.

Other work on fast sampling can be viewed as manual or automated methods to adjust samplers or diffusion processes for fast generation. Nichol & Dhariwal (2021); Kong & Ping (2021) describe methods to adjust a discrete time diffusion model trained on many timesteps into models that can sample in few timesteps. Watson et al. (2021) describe a dynamic programming algorithm to reduce the number of timesteps for a diffusion model in a way that is optimal for log likelihood. Chen et al. (2021); Saharia et al. (2021); Ho et al. (2021) train diffusion models over continuous noise levels and tune samplers post training by adjusting the noise levels of a few-step discrete time reverse diffusion process. Their method is effective in highly conditioned settings such as text-to-speech and image super-resolution. San-Roman et al. (2021) train a new network to estimate the noise level of noisy data and show how to use this estimate to speed up sampling.

Alternative specifications of the diffusion model can also lend themselves to fast sampling, such as modified forward and reverse processes (Nachmani et al., 2021; Lam et al., 2021) and training diffusion models in latent space (Vahdat et al., 2021).

| Method | Model evaluations | FID |
|---|---|---|
| Progressive Distillation (ours) | 1 | 9.12 |
| | 2 | 4.51 |
| | 4 | 3.00 |
| | 8 | 2.57 |
| Knowledge distillation (Luhman & Luhman, 2021) | 1 | 9.36 |
| DDIM (Song et al., 2021a) | 10 | 13.36 |
| | 20 | 6.84 |
| | 50 | 4.67 |
| | 100 | 4.16 |
| Dynamic step-size extrapolation + VP-deep | 48 | 82.42 |
| (Jolicoeur-Martineau et al., 2021) | 151 | 2.73 |
| | 180 | 2.44 |
| | 274 | 2.60 |
| | 330 | 2.56 |
| FastDPM (Kong & Ping, 2021) | 10 | 9.90 |
| | 20 | 5.05 |
| | 50 | 3.20 |
| | 100 | 2.86 |
| Improved DDPM respacing | 25 | 7.53 |
| (Nichol & Dhariwal, 2021), our reimplementation | 50 | 4.99 |
| LSGM (Vahdat et al., 2021) | 138 | 2.10 |

Table 2: Comparison of fast sampling results on CIFAR-10 for diffusion models in the literature.

## 7 DISCUSSION

We have presented *progressive distillation*, a method to drastically reduce the number of sampling steps required for high quality generation of images, and potentially other data, using diffusion models with deterministic samplers like DDIM (Song et al., 2020). By making these models cheaper to run at test time, we hope to increase their usefulness for practical applications, for which running time and computational requirements often represent important constraints.

In the current work we limited ourselves to setups where the student model has the same architecture and number of parameters as the teacher model: in future work we hope to relax this constraint and explore settings where the student model is smaller, potentially enabling further gains in test time computational requirements. In addition, we hope to move past the generation of images and also explore progressive distillation of diffusion models for different data modalities such as e.g. audio (Chen et al., 2021).

In addition to the proposed distillation procedure, some of our progress was realized through different parameterizations of the diffusion model and its training loss. We expect to see more progress in this direction as the community further explores this model class.

## REPRODUCIBILITY STATEMENT

We provide full details on model architectures, training procedures, and hyperparameters in Appendix E, in addition to our discussion in Section 5. In Algorithm 2 we provide fairly detailed pseudocode that closely matches our actual implementation, which is available in open source at `https://github.com/google-research/google-research/tree/master/diffusion_distillation`.

## ETHICS STATEMENT

In general, generative models can have unethical uses, such as fake content generation, and they can suffer from bias if applied to data sets that are not carefully curated. The focus of this paper specifically is on speeding up generative models at test time in order to reduce their computational demands; we do not have specific concerns with regards to this contribution.

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

## A  PROBABILITY FLOW ODE IN TERMS OF LOG-SNR

Song et al. (2021c) formulate the forward diffusion process in terms of an SDE of the form

$$d\mathbf{z} = f(\mathbf{z}, t)dt + g(t)dW, \tag{10}$$

and show that samples from this diffusion process can be generated by solving the associated *probability flow* ODE:

$$d\mathbf{z} = [f(\mathbf{z}, t) - \frac{1}{2}g^2(t)\nabla_z \log p_t(\mathbf{z})]dt, \tag{11}$$

where in practice $\nabla_z \log p_t(\mathbf{z})$ is approximated by a learned denoising model using

$$\nabla_z \log p_t(\mathbf{z}) \approx \frac{\alpha_t \hat{\mathbf{x}}_\theta(\mathbf{z}_t) - \mathbf{z}_t}{\sigma_t^2}. \tag{12}$$

Following Kingma et al. (2021) we have $f(\mathbf{z}, t) = \frac{d \log \alpha_t}{dt}\mathbf{z}_t$ and $g^2(t) = \frac{d\sigma_t^2}{dt} - 2\frac{d \log \alpha_t}{dt}\sigma_t^2$. Assuming a variance preserving diffusion process with $\alpha_t^2 = 1 - \sigma_t^2 = \text{sigmoid}(\lambda_t)$ for $\lambda_t = \log[\alpha_t^2/\sigma_t^2]$ (without loss of generality, see Kingma et al. (2021)), we get

$$f(\mathbf{z}, t) = \frac{d \log \alpha_t}{dt}\mathbf{z}_t = \frac{1}{2}\frac{d \log \alpha_\lambda^2}{d\lambda}\frac{d\lambda}{dt}\mathbf{z}_t = \frac{1}{2}(1 - \alpha_t^2)\frac{d\lambda}{dt}\mathbf{z}_t = \frac{1}{2}\sigma_t^2\frac{d\lambda}{dt}\mathbf{z}_t. \tag{13}$$

Similarly, we get

$$g^2(t) = \frac{d\sigma_t^2}{dt} - 2\frac{d \log \alpha_t}{dt}\sigma_t^2 = \frac{d\sigma_\lambda^2}{d\lambda}\frac{d\lambda}{dt} - \sigma_t^4\frac{d\lambda}{dt} = (\sigma_t^4 - \sigma_t^2)\frac{d\lambda}{dt} - \sigma_t^4\frac{d\lambda}{dt} = -\sigma_t^2\frac{d\lambda}{dt}. \tag{14}$$

Plugging these into the probability flow ODE then gives

$$d\mathbf{z} = [f(\mathbf{z}, t) - \frac{1}{2}g^2(t)\nabla_z \log p_t(\mathbf{z})]dt \tag{15}$$

$$= \frac{1}{2}\sigma_\lambda^2[\mathbf{z}_\lambda + \nabla_z \log p_\lambda(\mathbf{z})]d\lambda. \tag{16}$$

Plugging in our function approximation from Equation 12 gives

$$d\mathbf{z} = \frac{1}{2}\sigma_\lambda^2\left[\mathbf{z}_\lambda + \left(\frac{\alpha_\lambda \hat{\mathbf{x}}_\theta(\mathbf{z}_\lambda) - \mathbf{z}_\lambda}{\sigma_\lambda^2}\right)\right]d\lambda \tag{17}$$

$$= \frac{1}{2}[\alpha_\lambda \hat{\mathbf{x}}_\theta(\mathbf{z}_\lambda) + (\sigma_\lambda^2 - 1)\mathbf{z}_\lambda]d\lambda \tag{18}$$

$$= \frac{1}{2}[\alpha_\lambda \hat{\mathbf{x}}_\theta(\mathbf{z}_\lambda) - \alpha_\lambda^2\mathbf{z}_\lambda]d\lambda. \tag{19}$$

## B  DDIM IS AN INTEGRATOR OF THE PROBABILITY FLOW ODE

The DDIM update rule (Song & Ermon, 2020) is given by

$$\mathbf{z}_s = \frac{\sigma_s}{\sigma_t}[\mathbf{z}_t - \alpha_t \hat{\mathbf{x}}_\theta(\mathbf{z}_t)] + \alpha_s \hat{\mathbf{x}}_\theta(\mathbf{z}_t), \tag{20}$$

for $s < t$. Taking the derivative of this expression with respect to $\lambda_s$, assuming again a variance preserving diffusion process, and using $\frac{d\alpha_\lambda}{d\lambda} = \frac{1}{2}\alpha_\lambda \sigma_\lambda^2$ and $\frac{d\sigma_\lambda}{d\lambda} = -\frac{1}{2}\sigma_\lambda \alpha_\lambda^2$, gives

$$\frac{\mathbf{z}_{\lambda_s}}{d\lambda_s} = \frac{d\sigma_{\lambda_s}}{d\lambda_s}\frac{1}{\sigma_t}[\mathbf{z}_t - \alpha_t \hat{\mathbf{x}}_\theta(\mathbf{z}_t)] + \frac{d\alpha_{\lambda_s}}{d\lambda_s}\hat{\mathbf{x}}_\theta(\mathbf{z}_t) \tag{21}$$

$$= -\frac{1}{2}\alpha_s^2\frac{\sigma_s}{\sigma_t}[\mathbf{z}_t - \alpha_t \hat{\mathbf{x}}_\theta(\mathbf{z}_t)] + \frac{1}{2}\alpha_s\sigma_s^2\hat{\mathbf{x}}_\theta(\mathbf{z}_t). \tag{22}$$

Evaluating this derivative at $s = t$ then gives

$$\frac{\mathbf{z}_{\lambda_s}}{d\lambda_s}|_{s=t} = -\frac{1}{2}\alpha_\lambda^2[\mathbf{z}_\lambda - \alpha_\lambda \hat{\mathbf{x}}_\theta(\mathbf{z}_\lambda)] + \frac{1}{2}\alpha_\lambda\sigma_\lambda^2\hat{\mathbf{x}}_\theta(\mathbf{z}_\lambda) \tag{23}$$

$$= -\frac{1}{2}\alpha_\lambda^2[\mathbf{z}_\lambda - \alpha_\lambda \hat{\mathbf{x}}_\theta(\mathbf{z}_\lambda)] + \frac{1}{2}\alpha_\lambda(1 - \alpha_\lambda^2)\hat{\mathbf{x}}_\theta(\mathbf{z}_\lambda) \tag{24}$$

$$= \frac{1}{2}[\alpha_\lambda \hat{\mathbf{x}}_\theta(\mathbf{z}_\lambda) - \alpha_\lambda^2\mathbf{z}_\lambda]. \tag{25}$$

Comparison with Equation 19 now shows that DDIM follows the probability flow ODE up to first order, and can thus be considered as an integration rule for this ODE.

## C    EVALUATION OF INTEGRATORS OF THE PROBABILITY FLOW ODE

In a preliminary investigation we tried several numerical integrators for the probability flow ODE. As our model we used a pre-trained class-conditional 128x128 ImageNet model following the description in Ho et al. (2020). We tried a simple Euler integrator, RK4 (the "classic" 4th order Runge–Kutta integrator), and DDIM (Song et al., 2021a). In addition we compared to a Gaussian sampler with variance equal to the lower bound given by Ho et al. (2020). We calculated FID scores on just 5000 samples, hence our results in this experiment are not comparable to results reported in the literature. This preliminary investigation gave the results listed in Table 3 and identified DDIM as the best integrator in terms of resulting sample quality.

| Sampler | Number of steps | FID |
|---|---|---|
| Stochastic | 1000 | **13.35** |
| Euler | 1000 | 16.5 |
| RK4 | 1000 | 16.33 |
| DDIM | 1000 | 15.98 |
| Stochastic | 100 | 18.44 |
| Euler | 100 | 23.67 |
| RK4 | 100 | 18.94 |
| DDIM | 100 | **16.35** |

Table 3: Preliminary FID scores on $128 \times 128$ ImageNet for various integrators of the probability flow ODE, and compared against a stochastic sampler. Model specification and noise schedule follow Ho et al. (2020).

## D    EXPRESSION OF DDIM IN ANGULAR PARAMETERIZATION

We can simplify the DDIM update rule by expressing it in terms of $\phi_t = \arctan(\sigma_t/\alpha_t)$, rather than in terms of time $t$ or log-SNR $\lambda_t$, as we show here.

Given our definition of $\phi$, and assuming a variance preserving diffusion process, we have $\alpha_\phi = \cos(\phi)$, $\sigma_\phi = \sin(\phi)$, and hence $\mathbf{z}_\phi = \cos(\phi)\mathbf{x} + \sin(\phi)\epsilon$. We can now define the velocity of $\mathbf{z}_\phi$ as

$$\mathbf{v}_\phi \equiv \frac{d\mathbf{z}_\phi}{d\phi} = \frac{d\cos(\phi)}{d\phi}\mathbf{x} + \frac{d\sin(\phi)}{d\phi}\epsilon = \cos(\phi)\epsilon - \sin(\phi)\mathbf{x}. \tag{26}$$

Rearranging $\epsilon, \mathbf{x}, \mathbf{v}$, we then get

$$\sin(\phi)\mathbf{x} = \cos(\phi)\epsilon - \mathbf{v}_\phi \tag{27}$$

$$= \frac{\cos(\phi)}{\sin(\phi)}(\mathbf{z} - \cos(\phi)\mathbf{x}) - \mathbf{v}_\phi \tag{28}$$

$$\sin^2(\phi)\mathbf{x} = \cos(\phi)\mathbf{z} - \cos^2(\phi)\mathbf{x} - \sin(\phi)\mathbf{v}_\phi \tag{29}$$

$$(\sin^2(\phi) + \cos^2(\phi))\mathbf{x} = \mathbf{x} = \cos(\phi)\mathbf{z} - \sin(\phi)\mathbf{v}_\phi, \tag{30}$$

and similarly we get $\epsilon = \sin(\phi)\mathbf{z}_\phi + \cos(\phi)\mathbf{v}_\phi$.

Furthermore, we define the predicted velocity as

$$\hat{\mathbf{v}}_\theta(\mathbf{z}_\phi) \equiv \cos(\phi)\hat{\epsilon}_\theta(\mathbf{z}_\phi) - \sin(\phi)\hat{\mathbf{x}}_\theta(\mathbf{z}_\phi), \tag{31}$$

where $\hat{\epsilon}_\theta(\mathbf{z}_\phi) = (\mathbf{z}_\phi - \cos(\phi)\hat{\mathbf{x}}_\theta(\mathbf{z}_\phi))/\sin(\phi)$.

Rewriting the DDIM update rule in the introduced terms then gives

$$\mathbf{z}_{\phi_s} = \cos(\phi_s)\hat{\mathbf{x}}_\theta(\mathbf{z}_{\phi_t}) + \sin(\phi_s)\hat{\epsilon}_\theta(\mathbf{z}_{\phi_t}) \tag{32}$$

$$= \cos(\phi_s)(\cos(\phi_t)\mathbf{z}_{\phi_t} - \sin(\phi_t)\hat{\mathbf{v}}_\theta(\mathbf{z}_{\phi_t})) + \sin(\phi_s)(\sin(\phi_t)\mathbf{z}_{\phi_t} + \cos(\phi_t)\hat{\mathbf{v}}_\theta(\mathbf{z}_{\phi_t})) \tag{33}$$

$$= [\cos(\phi_s)\cos(\phi_t) - \sin(\phi_s)\sin(\phi_t)]\mathbf{z}_{\phi_t} + [\sin(\phi_s)\cos(\phi_t) - \cos(\phi_s)\sin(\phi_t)]\hat{\mathbf{v}}_\theta(\mathbf{z}_{\phi_t}). \tag{34}$$

Finally, we use the trigonometric identities

$$\cos(\phi_s)\sin(\phi_t) - \sin(\phi_s)\cos(\phi_t) = \cos(\phi_s - \phi_t) \tag{35}$$
$$\sin(\phi_s)\cos(\phi_t) - \cos(\phi_s)\sin(\phi_t) = \sin(\phi_s - \phi_t), \tag{36}$$

to find that

$$\mathbf{z}_{\phi_s} = \cos(\phi_s - \phi_t)\mathbf{z}_{\phi_t} + \sin(\phi_s - \phi_t)\hat{\mathbf{v}}_\theta(\mathbf{z}_{\phi_t}). \tag{37}$$

or equivalently

$$\mathbf{z}_{\phi_t - \delta} = \cos(\delta)\mathbf{z}_{\phi_t} - \sin(\delta)\hat{\mathbf{v}}_\theta(\mathbf{z}_{\phi_t}). \tag{38}$$

Viewed from this perspective, DDIM thus evolves $\mathbf{z}_{\phi_s}$ by moving it on a circle in the $(\mathbf{z}_{\phi_t}, \hat{\mathbf{v}}_{\phi_t})$ basis, along the $-\hat{\mathbf{v}}_{\phi_t}$ direction. The relationship between $\mathbf{z}_{\phi_t}, \mathbf{v}_t, \alpha_t, \sigma_t, \mathbf{x}, \epsilon$ is visualized in Figure 5.

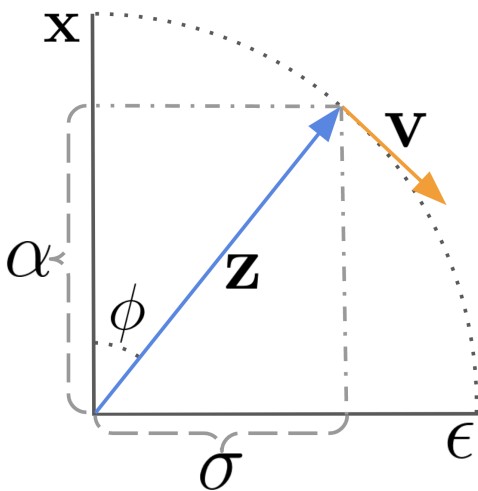

Figure 5: Visualization of reparameterizing the diffusion process in terms of $\phi$ and $\mathbf{v}_\phi$.

## E  SETTINGS USED IN EXPERIMENTS

Our model architectures closely follow those described by Dhariwal & Nichol (2021). For $64 \times 64$ ImageNet we use their model exactly, with 192 channels at the highest resolution. All other models are slight variations with different hyperparameters.

For CIFAR-10 we use an architecture with a fixed number of channels at all resolutions of 256. The model consists of a UNet that internally downsamples the data twice, to $16 \times 16$ and to $8 \times 8$. At each resolution we apply 3 residual blocks, like described by Dhariwal & Nichol (2021). We use single-headed attention, and only apply this at the $16 \times 16$ and $8 \times 8$ resolutions. We use dropout of 0.2 when training the original model. No dropout is used during distillation.

For LSUN we use a model similar to that for ImageNet, but with a reduced number of 128 channels at the $64 \times 64$ resolution. Compared to ImageNet we have an additional level in the UNet, corresponding to the input resolution of $128 \times 128$, which we process using 3 residual blocks with 64 channels. We only use attention layers for the resolutions of $32 \times 32$ and lower.

For CIFAR-10 we take the output of the model to represent a prediction of $\mathbf{x}$ directly, as discussed in Section 4. For the other data sets we used the combined prediction of $(\mathbf{x}, \epsilon)$ like described in that section also. All original models are trained with Adam with standard settings (learning rate of $3 * 10^{-4}$), using a parameter moving average with constant 0.9999 and very slight decoupled weight decay (Loshchilov & Hutter, 2017) with a constant of 0.001. We clip the norm of gradients to a global norm of 1 before calculating parameter updates. For CIFAR-10 we train for 800k parameter updates, for ImageNet we use 550k updates, and for LSUN we use 400k updates. During distillation we train for 50k updates per iteration, except for the distillation to 2 and 1 sampling steps, for which we use 100k updates. We linearly anneal the learning rate from $10^{-4}$ to zero during each iteration.

We use a batch size of 128 for CIFAR-10 and 2048 for the other data sets. We run our experiments on TPUv4, using 8 TPU chips for CIFAR-10, and 64 chips for the other data sets. The total time required to first train and then distill a model varies from about a day for CIFAR-10, to about 5 days for ImageNet.

## F   STOCHASTIC SAMPLING WITH DISTILLED MODELS

Our progressive distillation procedure was designed to be used with the DDIM sampler, but the resulting distilled model could in principle also be used with a stochastic sampler. Here we evaluate a distilled model for 64x64 ImageNet using the optimized stochastic sampler also used in Section 5.2. The results are presented in Figure 6.

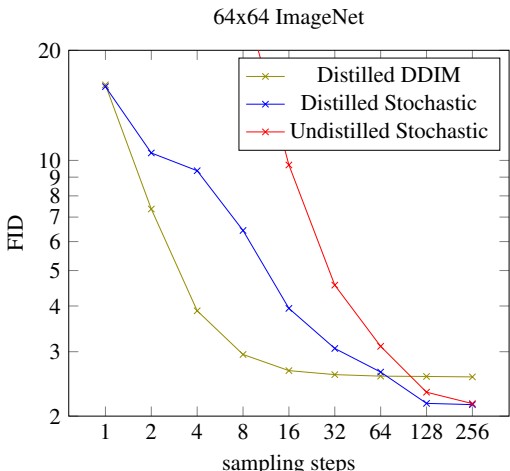

Figure 6: FID of generated samples from distilled and undistilled models, using DDIM or stochastic sampling. For the stochastic sampling results we present the best FID obtained by a grid-search over 11 possible noise levels, spaced log-uniformly between the upper and lower bound on the variance as derived by Ho et al. (2020). The performance of the distilled model with stochastic sampling is found to lie in between the undistilled original model with stochastic sampling and the distilled DDIM sampler: For small numbers of sampling steps the DDIM sampler performs better with the distilled model, for large numbers of steps the stochastic sampler performs better.

## G   DERIVATION OF THE DISTILLATION TARGET

The key difference between our progressive distillation algorithm proposed in Section 3 and the standard diffusion training procedure is in how we determine the target value for our denoising model. In standard diffusion training, the target for denoising is the clean data $\mathbf{x}$. In progressive distillation it is the value $\tilde{\mathbf{x}}$ the student denoising model would need to predict in order to match the teacher model when sampling. Here we derive what this target needs to be.

Using notation $t' = t - 0.5/N$ and $t'' = t - 1/N$, when training a student with $N$ sampling steps, we have that the teacher model samples the next set of noisy data $\mathbf{z}_{t''}$ given the current noisy data $\mathbf{z}_t$ by taking two steps of DDIM. The student tries to sample the same value in only one step of DDIM. Denoting the student denoising prediction by $\tilde{\mathbf{x}}$, and its one-step sample by $\tilde{\mathbf{z}}_{t''}$, application of the DDIM sampler (see equation 8), gives:

$$\tilde{\mathbf{z}}_{t''} = \alpha_{t''}\tilde{\mathbf{x}} + \frac{\sigma_{t''}}{\sigma_t}(\mathbf{z}_t - \alpha_t\tilde{\mathbf{x}}). \tag{39}$$

In order for the student sampler to match the teacher sampler, we must set $\tilde{\mathbf{z}}_{t''}$ equal to $\mathbf{z}_{t''}$. This gives

$$\tilde{\mathbf{z}}_{t''} = \alpha_{t''}\tilde{\mathbf{x}} + \frac{\sigma_{t''}}{\sigma_t}(\mathbf{z}_t - \alpha_t\tilde{\mathbf{x}}) = \mathbf{z}_{t''} \tag{40}$$

$$= \left(\alpha_{t''} - \frac{\sigma_{t''}}{\sigma_t}\alpha_t\right)\tilde{\mathbf{x}} + \frac{\sigma_{t''}}{\sigma_t}\mathbf{z}_t = \mathbf{z}_{t''} \tag{41}$$

$$\left(\alpha_{t''} - \frac{\sigma_{t''}}{\sigma_t}\alpha_t\right)\tilde{\mathbf{x}} = \mathbf{z}_{t''} - \frac{\sigma_{t''}}{\sigma_t}\mathbf{z}_t \tag{42}$$

$$\tilde{\mathbf{x}} = \frac{\mathbf{z}_{t''} - \frac{\sigma_{t''}}{\sigma_t}\mathbf{z}_t}{\alpha_{t''} - \frac{\sigma_{t''}}{\sigma_t}\alpha_t} \tag{43}$$

In other words, if our student denoising model exactly predicts $\tilde{\mathbf{x}}$ as defined in equation 43 above, then the one-step student sample $\tilde{\mathbf{z}}_{t''}$ is identical to the two-step teacher sample $\mathbf{z}_{t''}$. In order to have our student model approximate this ideal outcome, we thus train it to predict $\tilde{\mathbf{x}}$ from $\mathbf{z}_t$ as well as possible, using the standard squared error denoising loss (see Equation 9).

Note that this possibility of matching the two-step teacher model with a one-step student model is unique to deterministic samplers like DDIM: the composition of two standard stochastic DDPM sampling steps (Equation 5) forms a non-Gaussian distribution that falls outside the family of Gaussian distributions that can be modelled by a single DDPM student step: A multi-step stochastic DDPM sampler can thus not be distilled into a few-step sampler without some loss in fidelity. This is in contrast with the deterministic DDIM sampler: here both the two-step DDIM teacher update and the one-step DDIM student update represent deterministic mappings implemented by a neural net, which is why the student is able to accurately match the teacher.

Finally, note that we do lose something during the progressive distillation process: while the original model was trained to denoise $\mathbf{z}_t$ for any given continuous time $t$, the distilled student models are only ever evaluated on a small discrete set of times $t$. The student models thus lose generality as distillation progresses. At the same time, it's this loss of generality that allows the student models to free up enough modeling capacity to accurately match the teacher model without increasing their model size.

## H  ADDITIONAL RANDOM SAMPLES

In this section we present additional random samples from our diffusion models obtained through progressive distillation. We show samples for distilled models taking 256, 4, and 1 sampling steps. All samples are uncurated.

As explained in Section 3, our distilled samplers implement a deterministic mapping from input noise to output samples (also see Appendix G). To facilitate comparison of this mapping for varying numbers of sampling steps, we generate all samples using the same random input noise, and we present the samples side-by-side. As these samples show, the mapping is mostly preserved when moving from many steps to a single step: The same input noise is mapped to the same output image, with a slight loss in image quality, as the number of steps is reduced. Since the mapping is preserved while reducing the number of steps, our distilled models also preserve the excellent sample diversity of diffusion models (see e.g. Kingma et al. (2021)).

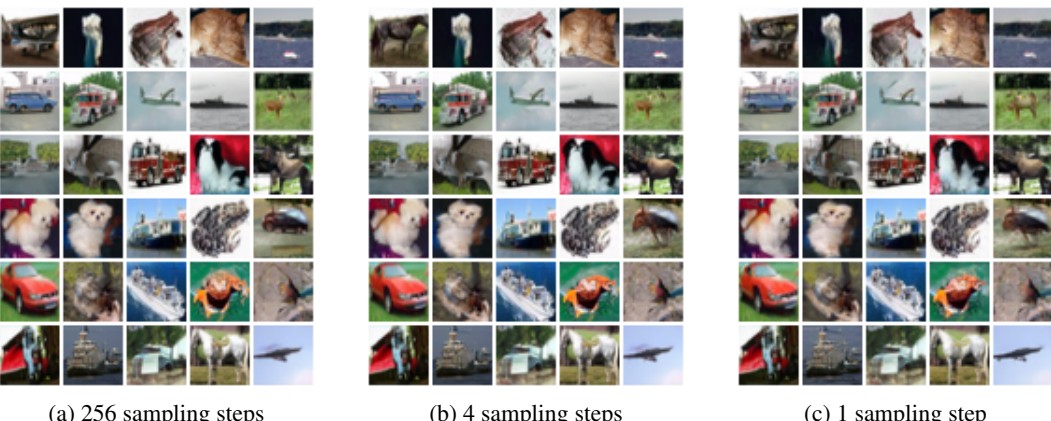

(a) 256 sampling steps    (b) 4 sampling steps    (c) 1 sampling step

Figure 7: Random samples from our distilled CIFAR-10 models, for fixed random seed and for varying number of sampling steps.

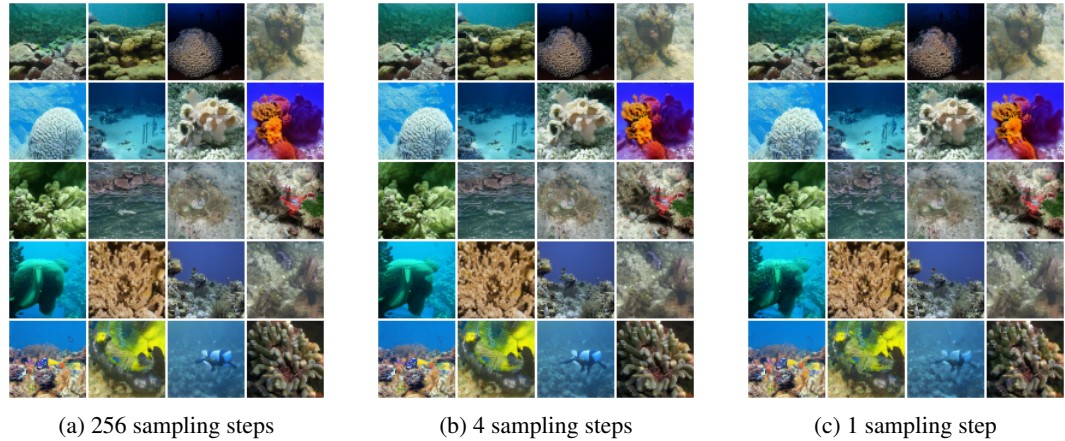

(a) 256 sampling steps    (b) 4 sampling steps    (c) 1 sampling step

Figure 8: Random samples from our distilled $64 \times 64$ ImageNet models, conditioned on the 'coral reef' class, for fixed random seed and for varying number of sampling steps.

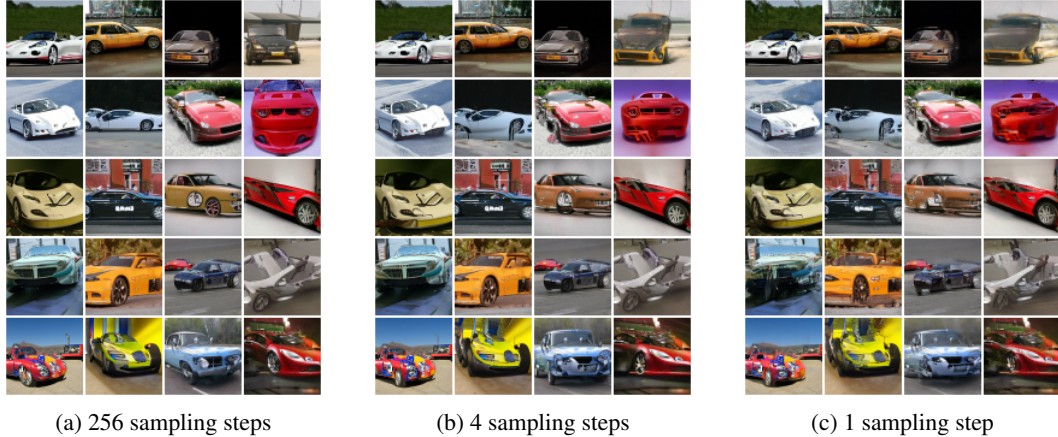

(a) 256 sampling steps    (b) 4 sampling steps    (c) 1 sampling step

Figure 9: Random samples from our distilled $64 \times 64$ ImageNet models, conditioned on the 'sports car' class, for fixed random seed and for varying number of sampling steps.

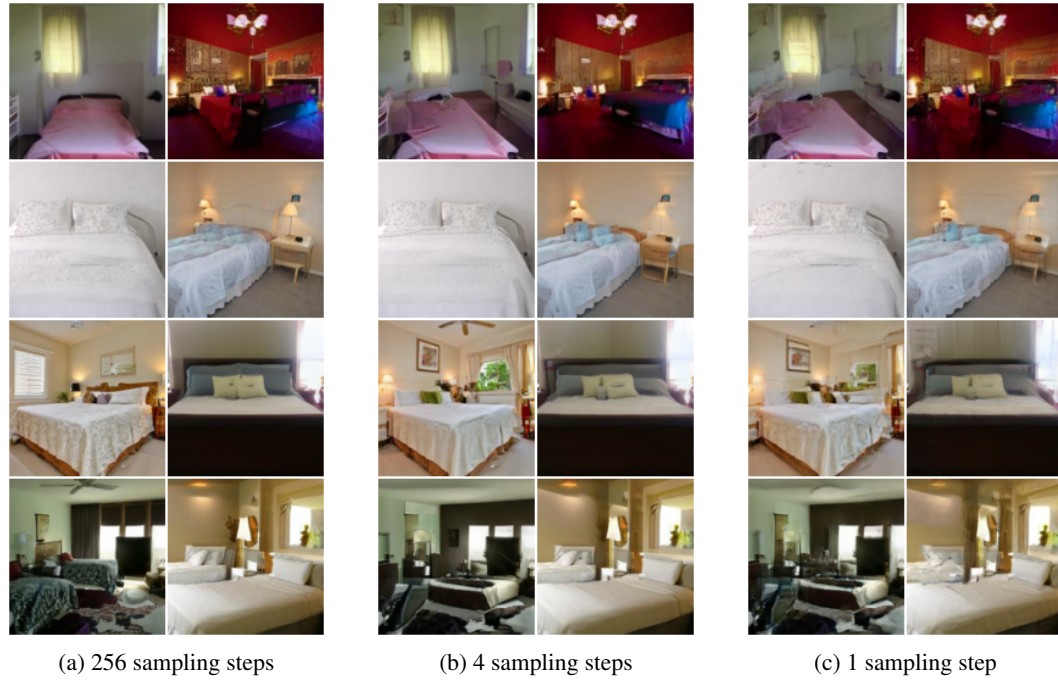

| (a) 256 sampling steps | (b) 4 sampling steps | (c) 1 sampling step |

Figure 10: Random samples from our distilled LSUN bedrooms models, for fixed random seed and for varying number of sampling steps.

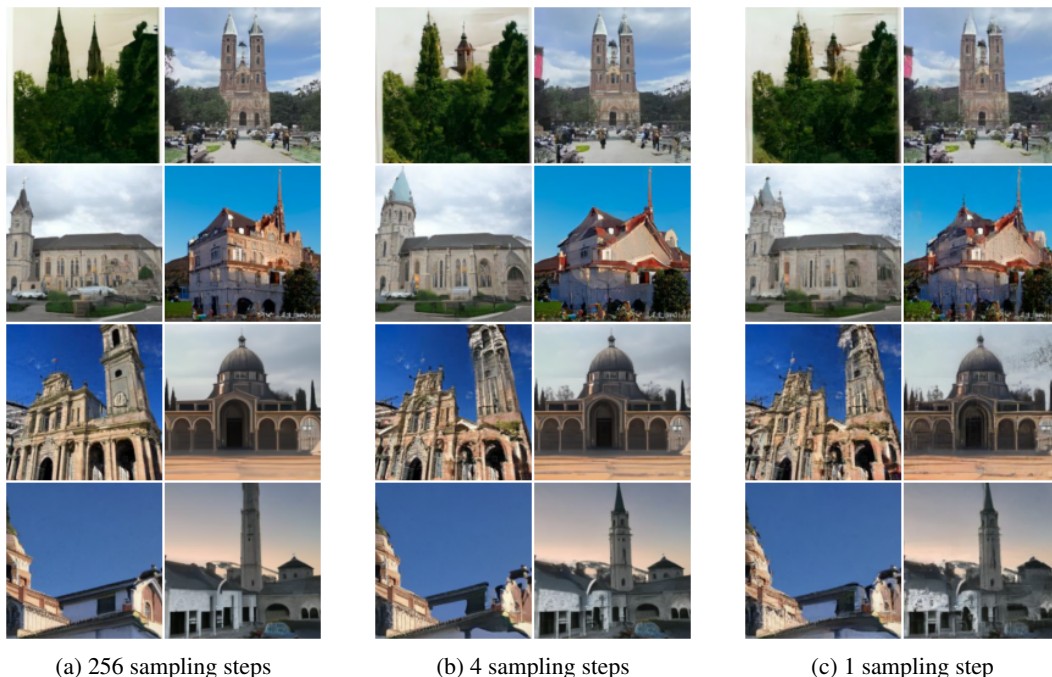

| (a) 256 sampling steps | (b) 4 sampling steps | (c) 1 sampling step |

Figure 11: Random samples from our distilled LSUN church-outdoor models, for fixed random seed and for varying number of sampling steps.

# I  ABLATION WITH FASTER DISTILLATION SCHEDULES

In order to further reduce the computational requirements for our progressive distillation approach, we perform an ablation study on CIFAR-10, where we decrease the number of parameter updates we use to train each new student model. In Figure 12 we present results for taking 25 thousand, 10 thousand, or 5 thousand optimization steps, instead of the 50 thousand we suggested in Section 3. As the results show, we can drastically decrease the number of optimization steps taken, and still get very good performance when using $\geq 4$ sampling steps. When taking very few sampling steps, the loss in performance becomes more pronounced when training the student for only a short time.

In addition to just decreasing the number of parameter updates, we also experiment with a schedule where we train each student on 4 times fewer sampling steps than its teacher, rather than the 2 times we propose in Section 3. Here the denoising target is still derived from taking 2 DDIM steps with the teacher model as usual, since taking 4 teacher steps would negate most of the computational savings. As Figure 12 shows, this does not work as well: if the computational budget is limited, it's better to take fewer parameter updates per halving of the number of sampling steps then to skip distillation iterations altogether.

In Figure 13 we show the results achieved with a faster schedule for the ImageNet and LSUN datasets. Here also, we achieve excellent results with a faster distillation schedule.

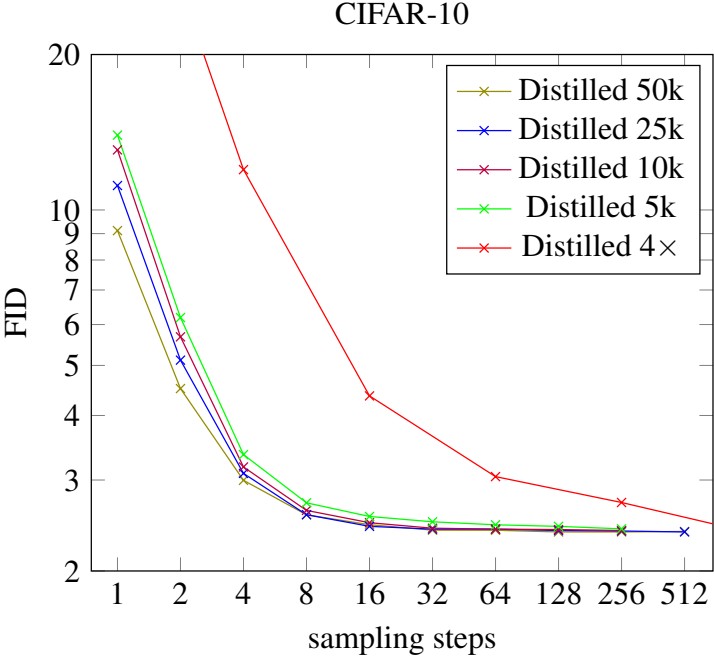

Figure 12: Comparing our proposed schedule for progressive distillation taking 50k parameter updates to train a new student every time the number of steps is halved, versus fast sampling schedules taking fewer parameter updates (25k, 10k, 5k), and a fast schedule dividing the number of steps by 4 for every new student instead of by 2. All reported numbers are averages over 4 random seeds. For each schedule we selected the optimal learning rate from $[5e^{-5}, 1e^{-4}, 2e^{-4}, 3e^{-4}]$.

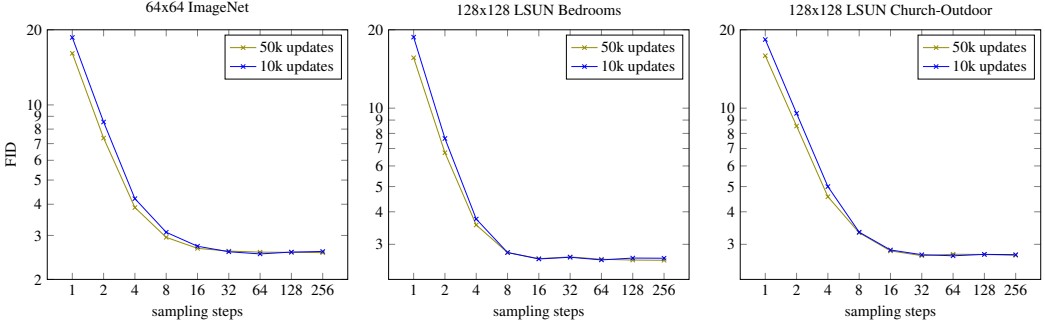

Figure 13: Comparing our proposed schedule for progressive distillation taking 50k parameter updates to train a new student every time the number of steps is halved, versus a fast sampling schedule taking 10k parameter updates. For each reported number of steps we selected the optimal learning rate from $[5e^{-5}, 1e^{-4}, 2e^{-4}, 3e^{-4}]$. Results are for a single random seed.

