# OpenReview forum: "Progressive Distillation for Fast Sampling of Diffusion Models"
_ICLR.cc/2022/Conference — ICLR 2022 Spotlight_

### Official Review · Reviewer_HZnL · 2021-10-29

**Correctness:** 4
**Technical Novelty And Significance:** 3
**Empirical Novelty And Significance:** 3
**Recommendation:** 8
**Confidence:** 4

**Main Review:**

**Strengths**:
- The paper successfully addresses a crucial downside of generative diffusion models, i.e., their slow sampling speed. This is a problem of very high relevance.
- The idea to use the deterministic DDIM [1] sampling scheme and distill 2 teacher steps into a single student step in each round of distillation is interesting and novel. I think it makes the distillation process easier compared to the situation where we would have to deal with a fully stochastic sampler during distillations. The overall algorithm is relatively straight-forward, which is good.
- The paper demonstrates strong empirical results, outperforming all previous works that aimed specifically to accelerate sampling from diffusion models. In particular, reaching FID scores $\leq3.0$ in as few as 4 steps on CIFAR-10 is a very strong result.

**Weaknesses**:
- The work tackles image generation and aims purely at reaching high FID in as few as possible sampling steps. If our only goal is to reach high perceptual quality as measured by FID with quick sampling, then one could also simply use state-of-the-art GANs [2], which reach FID scores $\leq3.0$ in a single shot (on CIFAR-10). I think that one of the crucial advantages of generative diffusion models is that they reach particularly strong diversity, do not easily drop modes, and better cover the full data distribution in their standard formulation (i.e. no distillation), unlike GANs, which are known to suffer from mode dropping and related challenges. It is unclear whether these advantages of diffusion models are preserved or lost in the proposed distillation process. This aspect is not discussed in the paper and no experiments to investigate this aspect are presented. This suggests the following: (a) Is it possible to somehow measure the likelihood of held-out validation data under the final distilled model? (b) As I assume that this may be difficult, could we calculate standard recall metrics [3] and show that distribution coverage is not significantly decreased in the distillation (and maybe also compare to GANs)? (c) To provide some further evidence towards diversity preservation and mode coverage, we could also run 2D toy experiments on a mixture of Gaussians and show that the distilled model still faithfully covers all modes (standard GANs often tend to miss modes in such settings).
- The explanation why $\epsilon$-prediction is not well-suited for the distillation approach makes sense. However, the proposed solutions, this is, the different proposed model parametrizations and loss weightings, seem to come primarily from trial-and-error and aren't overly well motivated. Is there anything we can say about which parametrizations and loss weightings should be optimal with respect to certain criteria? Can we provide more insights here?
- I sort of agree with the intuition that using the DDIM framework for the distillation protocol makes sense. However, what would happen if we did a similar distillation while using the standard stochastic DDPM sampling? Would this still be possible? A more thorough explanation or an experiment would be very interesting to clearly show that it would not work and provide a more solid motivation for choosing DDIM-based distillation.
- The method is evaluated on image generation only. Will the distillation approach also work for diffusion models in other domains? What could be additional challenges that may potentially occur? In fact, I think it would be very promising to also apply the method on other data types, because in non-image domains there is less "competition" from the GAN literature.

**Additional Questions and Suggestions**:
- In the interest of a clean presentation, I would suggest to properly introduce the $\epsilon$ in equation (2).
- In cection 3: "By inverting a single step of DDIM, we then calculate the value the student model would need to predict in order to step from $z_t$ to $z_{t-1/N}$ in a single step." seems to correspond to the operation at "Teacher $\hat{x}$ target" in algorithm 2. I would consider explaining this in a bit more detail with more background and explicitly derive the expression for $\tilde{x}$ in algorithm 2. I think it could be made clearer and easier to follow and this is one of the most central operations in the proposed algorithm.
- If we distill the slow iterative sampling from diffusion models into these few-step samplers, one may intuitively expect that we pay a price in terms of requiring bigger network capacity to make up for the fewer synthesis steps. This does not seem to be the case, however. Why not? Note that this question is also related to the first point under *Weaknesses* above (i.e., maybe we don't use a bigger model, but have to pay a price with regards to diversity and mode coverage?).


[1] Song et al. "Denoising Diffusion Implicit Models", 2020.

[2] Karras et al. "Training Generative Adversarial Networks with Limited Data", 2020.

[3] Kynkäänniemi et al. "Improved Precision and Recall Metric for Assessing Generative Models", 2019.

**Summary Of The Paper:**

Diffusion models have recently emerged as a promising class of generative models. Their main drawback is the slow, iterative sampling, which often requires hundreds or thousands of neural network calls. The paper proposes a method to progressively distill this slow iterative generation process into much faster synthesis relying on fewer denoising steps, while almost preserving generation quality as measured by metrics quantifying perceptual image quality. The method progressively reduces the number of sampling steps required by a factor of 2 in multiple rounds of distillation and relies on a teacher-student framework. The student model learns to model 2 steps of the teacher in a single step. The method leverages the deterministic synthesis scheme from Denoising Diffusion Implicit Models for that purpose. To get the method to work successfully, new score model parametrizations and score matching loss weightings are proposed. The quantitative results are promising, for example achieving FID scores $\leq3.0$ on CIFAR-10 in as few as 4 steps.

**Summary Of The Review:**

The distillation idea in itself isn't entirely novel, as it was proposed by [4], too. However, I certainly acknowledge that this work shows how to do it in an elegant, scalable and high-performance manner using a progressive protocol. Overall, the paper addresses an important problem and shows promising quantitative results. Hence, I am leaning towards recommending acceptance. That said, some weaknesses remain. I would be willing to raise my score, if the weak points were addressed in a satisfactory manner.

[4] Luhmann and Luhmann, "Knowledge Distillation in Iterative Generative Models for Improved Sampling Speed", 2021.

---

> ### Author Response · Authors · 2021-11-18
> **author response**
>
> Thank you for your thorough review. We've just uploaded a new version of the paper in which we incorporated a number of your comments and suggestions. We plan to update again on Monday and will aim to incorporate anything we missed this time. Below we respond to your comments point by point.
>
> 1) *Do the distilled models preserve the sample diversity of the original diffusion model?*
>
> In our updated version of the paper (particularly appendix H) we show samples side-by-side for the various models produced during the progressive distillation process, i.e. the samples produced at 256 sampling steps (after almost no distillation), at 4 steps (heavy distillation), and at 1 step (maximum distillation). What's new in this version of the paper is that those samples all use the same random input noise / same random seed: This allows us to easily compare the mapping from input noise to output image that is implemented by the model at various stages of the distillation process. Judging by these side-by-side samples, we can say that the mapping is largely preserved and does not decrease in diversity as we progress with distillation; the noise inputs are largely mapped to the same images, and it is mostly the quality of those images that deteriorates as we take fewer sampling steps, not their content or diversity.
>
> > Is it possible to somehow measure the likelihood of held-out validation data under the final distilled model?
>
> Unfortunately not. Even the original undistilled model, when combined with the finite-step deterministic DDIM sampler, is not guaranteed to implement an invertible mapping between arbitrary noise and arbitrary images, and hence is not guaranteed to assign non-zero probability to any particular test image. Rather than thinking of the finite-step deterministic sampler itself as the generative model, we therefore prefer to think of it as an approximate sampler of a continuous-time stochastic diffusion process like the one analyzed in [1] and [2]. These models have excellent test-set likelihood, and all our analyses suggest our approximate sampler accurately reproduces their sampling distribution and sample diversity.
>
> > could we calculate standard recall metrics [3] and show that distribution coverage is not significantly decreased in the distillation
>
> We're currently working on implementing the recall metrics you suggest in our codebase, but are not sure we'll have results before the rebuttal deadline.
>
> 2) *Is there anything we can say about which parametrizations and loss weightings should be optimal with respect to certain criteria? Can we provide more insights here?*
>
> Regarding parameterization: Our main objective here was to find a parameterization that is stable in the signal-to-noise ratio, i.e. for which the sample iterates vary with the output of the neural network in equal amounts throughout the diffusion process from low SNR to high SNR. As we show in Appendix D, the v-parameterization is the unique parameterization that achieves this completely. We now also clearly present this insight in the main paper. From the empirical side, however, we find that approximately-stable parameterizations such as the joint (x, epsilon) parameterization work just as well.
>
> Regarding loss weighting: Different loss weightings emphasize different aspects of the data. When the objective is to maximize likelihood, [1] and [2] show that the optimal weighting is flat in the SNR. For optimizing perceptual quality no such result exists, and choosing the loss weighting is thus an entirely empirical exercise. By disentangling the parameterization from the loss weighting, and running an ablation with respect to both, we hope to offer some insight into the possible choices, but we cannot claim to give a definitive best answer here.
>
> ... continuing in a new response below due to the character limit
>
>
> References:
> [1] *Variational Diffusion Models*, Kingma et al. NeurIPS 2021. (https://papers.nips.cc/paper/2021/file/b578f2a52a0229873fefc2a4b06377fa-Paper.pdf)
> [2] *Maximum Likelihood Training of Score-Based Diffusion Models*, Song et al. NeurIPS 2021 (https://papers.nips.cc/paper/2021/file/0a9fdbb17feb6ccb7ec405cfb85222c4-Paper.pdf)

---

> > ### Author Response · Authors · 2021-11-18
> > **continuation**
> >
> > 3) *What would happen if we did a similar distillation while using the standard stochastic DDPM sampling? Would this still be possible? A more thorough explanation or an experiment would be very interesting.*
> >
> > We now devote some more discussion to this in the main part of the paper, as well as in Appendix F.
> > Some points:
> > * The composition of two standard stochastic DDPM sampling steps forms a non-Gaussian distribution that falls outside the family of Gaussian distributions that can be modelled by a single DDPM student step: A multi-step stochastic DDPM sampler can thus not be distilled in a few-step sampler without some loss in fidelity. This is in contrast with the deterministic DDIM sampler: here both the two-step DDIM teacher update and the one-step DDIM student update represent deterministic mappings implemented by a neural net: it is thus conceivable that the student could accurately match the teacher.
> > * The DDIM two-step sample update is deterministic and fully known given the noisy data z_t: This allows the student to model it very sharply, which enables it to progress more quickly during sampling compared to the teacher model. In contrast, in the stochastic case we would be denoising towards a mixture of possible samples, which would produce blurry predictions.
> > * Although distillation with the stochastic DDPM sampler does not seem possible, it *is* possible to evaluate the distilled model with the stochastic sampler. We have now added this experiment in Appendix F of the paper.
> >
> >
> > 4) *Will the distillation approach also work for diffusion models in other domains?*
> >
> > This is a great suggestion, and we're currently working hard to apply these models to other data modalities like audio and video, with encouraging initial results. However, we do not expect to be able to present those results already in the current paper.
> >
> >
> > 5) *Additional suggestions*
> >
> > > In the interest of a clean presentation, I would suggest to properly introduce the $\epsilon$ in equation (2).
> >
> > We now properly introduce epsilon when it is first used.
> >
> > > By inverting a single step of DDIM [...] I would consider explaining this in a bit more detail with more background and explicitly derive the expression for $\tilde{\mathbf{x}}$ in algorithm 2.
> >
> > We have now added this explanation and derivation in Appendix G.
> >
> > >  One may intuitively expect that we pay a price in terms of requiring bigger network capacity to make up for the fewer synthesis steps.
> > This does not seem to be the case, however. Why not?
> >
> > The original sampler takes many steps to produce its output, but this does not necessarily mean that it therefore implements a highly complex mapping that could not be represented in fewer steps. In fact, the original model is trained to denoise from any continuous noise level between time t=0 and t=1, while the student model is only ever evaluated on a small discrete set of specific time points corresponding to a small number of sampling steps: The task of the student model is thus easier in that respect, compared to the teacher model. In other words, the *price we pay* is that during distillation the student model loses this ability to denoise from arbitrary noise levels that was present in the original model.

---

> > > ### Comment · Reviewer_HZnL · 2021-11-19
> > > **Thanks you for reply**
> > >
> > > Thank you very much for the reply and the additional experiments. My main questions have been answered. I appreciate the experiment that is presented in Appendix H. I agree that this is evidence for diversity preservation. I would suggest to include a discussion about this directly in Appendix H. However, I would also still strongly suggest the authors to include quantitative results on recall metrics in the final version, as promised.
> > >
> > > The explanation regarding the difficulty of using the standard stochastic DDPM instead of DDIM also makes sense. Similarly, the explanation why the distilled model does not necessarily require more capacity seems plausible to me. I would suggest the authors to include such a discussion directly in the paper or Appendix.
> > >
> > > Overall, my main concerns have largely been eliminated. I am therefore raising my score from 6 to 8.

---

> > > > ### Author Response · Authors · 2021-11-22
> > > > **thanks**
> > > >
> > > > Thanks for raising your score.
> > > >
> > > > We have just updated the paper again and we have included the suggested discussion in Appendices H and G. In the final version of the paper we will also include the promised recall metrics.

---

### Official Review · Reviewer_orkh · 2021-11-01

**Correctness:** 3
**Technical Novelty And Significance:** 3
**Empirical Novelty And Significance:** 3
**Recommendation:** 8
**Confidence:** 4

**Details Of Ethics Concerns:**

The ethics concerns have been discussed in the paper: since its main goal is to reduce the sampling time of diffusion models, it does not introduce new concerns on the top of existing diffusion models.

**Main Review:**

Strengths:

(1) Overall, the presentation is clear and easy to understand, though there exist some minor issues with typos and confusing sentences (see comments below).

(2) The proposed progressive distillation is novel and interesting. Though the distillation idea to reduce the sampling time of diffusion models is not new, doing so in a progressive way seems to be more effective and efficient (which has been clearly discussed in the paper).

(3) The new parameterizations and training losses of diffusion models have been well motivated and explained to accommodate the proposed progressive distillation.

(4) Experiments have well supported the main claims and the effectiveness of the proposed method in reducing sampling time. In particular, the comparison with previous fast sampling methods on CIFAR-10 demonstrates that it largely outperforms these strong methods regarding FID scores with a few model evaluations.

Weaknesses:

(1) The progressive distillation process seems to need a much larger computational cost than many previous fast sampling methods, such as DDIM and DDPM respacing. As stated in the paper, its training budget is almost the same as training a diffusion model from scratch. I wonder how this concern can be addressed in practice?

(2) The main claim is that the method can reduce the number of model evaluations in sampling to as small as 4 or 8 steps while retaining high image quality. Currently, the highest resolution of considered datasets is 128x128. I wonder if the claim still holds for datasets with higher resolution? Does the resolution or complexity of the dataset impact the final steps of model evaluations?

(3) Minor issues in the writing. 1) Typos. For example, in the abstract, “as little as 4 steps” => “as few as 4 steps”. 2) Repeated references. For example, 	“Prafulla Dhariwal and Alex Nichol. Diffusion models beat GANs on image synthesis.”, and “Alex Nichol and Prafulla Dhariwal. Improved denoising diffusion probabilistic models. ”. 3) Confusing sentences. For example, what do you mean by saying “... unlike the original data point x, since multiple different data points x could conceivably have led to observing noisy data $z_t$”? Also, when saying “we found this to work slightly better than starting from a non-zero signal-to-noise ratio as used by e.g. Ho et al. (2020)”, does it refer to the undistilled sampler or the distilled sampler?

**Summary Of The Paper:**

This work studies the problem of improving the sampling speed of diffusion models, which is currently a timely and challenging topic. To tackle this problem, this work first proposes several new parameterizations of diffusion models for better stability along with a fast sampling, and then proposes a progressive distillation method that progressively distills two DDIM steps of teacher diffusion models to one DDIM step of student diffusion models. In the end, the method can reduce the number of model evaluations in sampling to as small as 4 or 8 steps while retaining high image quality. Experiments on CIFAR10, ImageNet (64x64), LSUN-bedrooms (128x128) and LSUN-Church (128x128) were conducted to show the effectiveness of the progrssive distillation in reducing the sampling time.


**Summary Of The Review:**

The proposed method is novel and effective in reducing the sample time of diffusion models, and experiments on several datasets well support its effectiveness and better performance than prior fast sampling methods. Still, I have some concerns about the method regarding computational cost and generalization to larger datasets. Thus, my initial rating is a weak accept. I’m willing to increase the score if my concerns can be addressed.

---

> ### Author Response · Authors · 2021-11-18
> **author response**
>
> Thank you for your comments. We've just uploaded a new version of the paper in which we incorporated your suggestions. We plan to update again on Monday, and will try to incorporate anything important we missed this time. Below we respond to your comments point by point:
>
>
> >  The progressive distillation process seems to need a much larger computational cost […] I wonder how this concern can be addressed in practice?
>
> Distilling a diffusion model after training indeed imposes additional computational cost at training time. We worked hard to keep the additional cost manageable, and below 1x the cost of training the original model. While indeed not inconsequential, this already appears to compare favourably to some other methods of speeding up diffusion models, such as the slower distillation approach of Luhmann and Luhmann, or the approach by Vahdat et al. where first a VAE needs to be trained before a diffusion model can be applied to the latents.
>
> Also, the added computational cost has to be considered in relation to how well the method works: Since our approach loses very little performance in distilling the original model to a small number of steps, the original model could be made smaller and cheaper, at the cost of slightly worse initial performance, while still ending up at a performance level that compares favourably to alternative methods that start out with the bigger model.
>
> Finding a practical tradeoff between performance and computational cost is an important problem in generative modelling, and we believe that progressive distillation could play an important role in finding better tradeoffs for many practical applications.
>
>
> > the method can reduce the number of model evaluations in sampling to as small as 4 or 8 steps while retaining high image quality. […] I wonder if the claim still holds for datasets with higher resolution?
>
> In the current paper we investigated generating images of resolution 32x32 to 128x128, and here we don’t see a clear relationship between resolution and the number of required model evaluations. In the literature, [1] and [2] studied the generation of higher resolution images using diffusion models, by first modeling a low resolution image and then upsampling to higher resolution using a second diffusion model. These studies found that their upsampling models require much fewer steps at sampling time compared to generating the original low resolution images. This suggests that higher resolution should not necessarily pose a problem, or alternatively that an efficient approach may be to train a distilled model for lower resolution images first, followed by another (distilled) model for upsampling.
>
>
> > Minor issues in the writing. Typos, repeated references, confusing sentences …
>
> Thanks for noticing these. We fixed all the specific issues you pointed out.
>
>
> > What do you mean by saying ``… unlike the original data point x, since multiple different data points x could conceivably have led to observing noisy data $z_t$ ”?
>
> What we meant here is to highlight the distinction between denoising towards the target $\tilde{\mathbf{x}}$ in progressive distillation, which is a deterministic function of the input $\mathbf{z}_t$ to the denoising model, and the original training objective of denoising to the clean training data $\mathbf{x}$, which cannot be uniquely determined given a set of noisy data $\mathbf{z}_t$, since multiple different data points $\mathbf{x}$ can give the same noisy observation $\mathbf{z}_t$ after random noise is added. Denoising to the fully determined target $\tilde{\mathbf{x}}$ rather than the uncertain original data $\mathbf{x}$ allows the student to learn to make sharper predictions, which is what allows for faster progress during sampling. We rewrote the section at the bottom of page 3 of the paper to hopefully explain this more clearly.
>
>
> > when saying “we found this to work slightly better than starting from a non-zero signal-to-noise ratio as used by e.g. Ho et al. (2020)”, does it refer to the undistilled sampler or the distilled sampler?
>
> This applies to both the distilled and undistilled sampler. We studied this most carefully for the undistilled sampler, where we did a grid search over starting signal-to-noise ratios and found smaller starting SNR to always be better. We added a bit more information to the discussion in the paper.
>
>
> References:
>
> [1] Saharia, Chitwan, et al. "Image super-resolution via iterative refinement." arXiv preprint arXiv:2104.07636 (2021).
>
> [2] Ho, Jonathan, et al. "Cascaded diffusion models for high fidelity image generation." arXiv preprint arXiv:2106.15282 (2021).

---

> > ### Comment · Reviewer_orkh · 2021-11-24
> > **Thanks for the response!**
> >
> > I thank the authors for the detailed response. Most of my concerns have been well addressed, in particular with the new experiments in Appendix I to discuss the impact of distillation schedule during training. Hence, I raised my rating from 6 to 8.

---

> ### Author Response · Authors · 2021-11-22
> **additional experiment to address concerns over computational cost**
>
> We have just revised the paper and have included a new experiment that will hopefully help alleviate any remaining concerns about the computational cost of the proposed approach. In Appendix I of the paper we now explore different distillation schedules that take fewer training steps for training each student model in the progressive distillation. TLDR: we can quite drastically reduce the computational budget for distillation, and still get very good results as long as we take >= 4 sampling steps. When using only 10% of the original budget, on CIFAR-10, we get an FID of 3.36 for 4 sampling steps, and 2.71 for 8 sampling steps. At this point, the computational cost of distillation starts to become negligible compared to training the original model.

---

### Official Review · Reviewer_2wPQ · 2021-11-02

**Correctness:** 4
**Technical Novelty And Significance:** 3
**Empirical Novelty And Significance:** 3
**Recommendation:** 8
**Confidence:** 3

**Main Review:**

Strengths
- An important topic given that diffusion models achieve very good results but that their sampling time is a barrier to wider adoption in practice.
- The idea of progressively distilling the model seems to be original and useful to avoid the need to evaluate the teacher model for many steps.
- Distillation to very few number of steps requires adaptions on the parameterization and loss weighting. Different plausible choices are presented and evaluated against each other. I particularly liked that this was evaluated on the original diffusion models without distillation to avoid confounding factors and to also provide insight on what works for these models themselves. Knowing that x-prediction works just as well is good to know since it can simplify training quite a bit.
- Overall, there are quite a few design choices for diffusion models which haven't been evaluated exhaustively and many of the results in this paper will help towards a better understanding on these choices (e.g. parameterization, weighting function, using signal-noise-ratio of 0 at t=1, different integrators evaluated in the supplementary).
- It also shows a nice interpretation of DDIM as an integrator of the probability flow ODE.
- Impressive results on retaining sample quality with very few steps, especially compared to previous approaches.
- The overall presentation is very clear, including the background on diffusion models, the proposed approach and difficulties, the algorithm, the results and the discussion of related works which also include very recent works.

Weaknesses
- The effect of distilling iteratively/progressively is not fully explored. Results in Tab. 2 suggest that when distilling to a single step model, the "one-step" distillation approach of Luhman&Luhman, 2021, performs similar to the presented approach. While potential advantages of not having to create a training datasets using the original model are mentioned in the Related Work Section, it would be nice to see a more detailed analysis, which, e.g. compares training costs between the two approaches. It would also be interesting to see the effect of choosing a different schedule for the progressive distillation, for example reducing the number of integration steps by a factor of four instead of two for each new student. This could have effects on both the time required for the distillation process as well as the quality achieved by the distilled models. In general, a bit more details on the time required for the distillation process and comparing that to other approaches might be useful.

Minor
- A bit contrary to what is claimed in the second paragraph of the introduction, [Dhariwal and Nichol, 2021b] also report good results with 25 steps in a class conditional setting (even though I agree with the general sentiment that tasks with less conditioning are more challenging).

**Summary Of The Paper:**

The paper presents an approach to improve sampling speed of diffusion models. The idea is to iteratively learn integrators for the probability flow ODE which, given a diffusion model, maps noise to data samples. In each iteration, the integrator is trained to integrate steps of the ODE over twice the length than the previous model. Thus, the approach resembles an iterative distillation process, where a faster student integrator is learned from a slower teacher integrator. After the student has finished learning, it becomes the teacher for the next distillation iteration.

Enabling this distillation requires changes to the parameterization and loss weighting, for which different, intuitive options are presented. Experiments evaluate the different design choices and compare the resulting models to other recent works on the topic, which demonstrates significant gains of the proposed approach in the regime where a small number of evaluations are used.

**Summary Of The Review:**

Given the impressive results obtained with diffusion models, the paper addresses an important and timely topic. The proposed approach significantly outperforms other methods in terms of quality achieved with distilled diffusion models at small number of evaluation steps, which, ultimately, is one of the main goals on this topic (even though the distillation process might be relatively expensive). The presentation is very clear and easy to follow and different possible design choices are well motivated and intuitive to understand. The different choices are evaluated in well designed experiments and comparisons are very convincing of the benefits of the presented approach and also include very recent works on the topic. The supplementary material also contains interesting material on different formulations and connections between different approaches. Thus, I recommend accepting this paper.

---

> ### Author Response · Authors · 2021-11-19
> **author response**
>
> Thank you for your review. We’ve updated the paper to include additional experiments and discussion, and plan to update again later to include anything we may have missed so far.
>
> Regarding your suggestion to further explore the effect of distilling progressively:
>
> > It would also be interesting to see the effect of choosing a different schedule for the progressive distillation, for example reducing the number of integration steps by a factor of four instead of two for each new student.
>
> We’ll run a *fast* version of our CIFAR experiment to experiment with 1: Reducing the number of integration steps by a factor of 4 for each new student, like you suggested. 2: Using half the number of parameter updates for distilling each student. We’ll include the results from this experiment in an appendix when we next update the paper (before the rebuttal deadline on Monday). If the results are interesting enough, we can run the same analysis for the other datasets for the camera ready version of the paper.
>
> > A bit contrary to what is claimed in the second paragraph of the introduction, [Dhariwal and Nichol, 2021b] also report good results with 25 steps in a class conditional setting (even though I agree with the general sentiment that tasks with less conditioning are more challenging).
>
> In addition to class-conditioning, Dhariwal and Nichol further condition their model by making use of an auxiliary classifier to guide the sampler. This indeed seems to enable speeding up the sampling procedure, and we now discuss this in the introduction of our paper as well. Note that this *classifier guidance* does come at the cost of reducing sample diversity, as e.g. shown in their Figure 5. In contrast, our method does not seem to impact sample diversity as evidenced by the FID and our visualised samples (also see our discussion with reviewer HZnL). Their classifier guidance method furthermore seems complementary to our distillation approach, and combining the two might be interesting to try in future work.
>
> Regarding you suggestion to provide a more complete comparison with Luhman & Luhman: We have expanded the discussion of this method a little bit in the related work section, as far a the available space allowed.

---

> > ### Author Response · Authors · 2021-11-22
> > **fast distillation experiment now included in revised paper**
> >
> > We've just revised the paper again and have included an experiment with fast distillation schedules in appendix I. TLDR: we can quite drastically reduce the computational budget for distillation, and still get very good results as long as we take >= 4 sampling steps. When using only 10% of the number of parameter updates, on CIFAR-10, we get an FID of 3.36 for 4 sampling steps, and 2.71 for 8 sampling steps. We compared reducing the number of integration steps by a factor of four instead of two for each new student against just reducing the number of training steps per student, and found the latter to be more effective.
> >
> > For the final version of the paper we'll also include the other data sets in this experiment.

---

### Official Review · Reviewer_3Sjb · 2021-11-08

**Correctness:** 4
**Technical Novelty And Significance:** 4
**Empirical Novelty And Significance:** 3
**Recommendation:** 8
**Confidence:** 4

**Main Review:**

## Strengths

1. Speeding up diffusion models is an important research direction, and achieving competitive performance with as few as 4 iterations is a remarkable achievement, which I expect to engender major impacts in the community.
2. Design choices like different parameterization of the score model, and different weighting functions in the score matching loss are clearly discussed. Their impact on model performance is also investigated rigorously in Table 1.
3. I especially like the discussion of DDIM as a numerical integrator for the probability flow ODE, and the experimental study on the performance of different numerical ODE solvers in the appendix.
4. Experiments demonstrate clear advantage over competing methods. It is surprising that even one iteration of the method is already sufficient for generating realistic images.

## Weaknesses

This paper is mostly focused on the variance preserving diffusion process. It is unclear how the distillation approached introduced in this paper can be applied to other SDEs such as VE SDEs and subVP SDEs, as well as how to choose the weighting function and score function parameterization in those cases.

**Summary Of The Paper:**

This work proposes to speed up diffusion models by progressively distilling a score network with deterministic dynamics. Authors report significant improvement of sample quality using very few iterations. Authors also provide a proof that DDIM is a special numerical integrator for the probability flow ODE, connecting two existing approaches. In addition, empirical studies on the weighting function and parameterization of score functions are also discussed.

**Summary Of The Review:**

Well written paper with impressive experimental results and clear description of design choices.

---

> ### Author Response · Authors · 2021-11-19
> **author response**
>
> Thank you for your review. We’ve updated the paper to include additional experiments and discussion, and plan to update again on Monday to include anything we may have missed so far.
>
> Regarding your comment on VE SDEs and subVP SDEs:
>
> > This paper is mostly focused on the variance preserving diffusion process. It is unclear how the distillation approach introduced in this paper can be applied to other SDEs such as VE SDEs and subVP SDEs, as well as how to choose the weighting function and score function parameterisation in those cases.
>
> Here, we would like to refer to the discussion on SDE specifications in [1], specifically section 5.1 and Appendix G: there they show that the various diffusion specifications, including VP SDE, VE SDE and subVP SDE are all equivalent, in the sense that they define the same generative model up to rescaling of the noisy latents $\mathbf{z}_t$. Our proposed distillation approach, as well as our recommendations about weighting function and score function parameterisation, can thus be applied to these other SDE specifications by converting between the VP-SDE and the other SDEs using the rescaling defined by Kingma et al., Appendix G. We’ve now also added this explanation and reference to the top of Section 4 of our paper.
>
> Reference: [1] *Variational Diffusion Models*, Kingma et al. NeurIPS 2021. (https://papers.nips.cc/paper/2021/file/b578f2a52a0229873fefc2a4b06377fa-Paper.pdf)

---

### Public Comment · ~Tim_Salimans1 · 2022-03-23
**code and checkpoints have been released!**

https://github.com/google-research/google-research/tree/master/diffusion_distillation

---

### Decision · Program_Chairs · 2022-01-20

**Decision:**

Accept (Spotlight)

**Comment:**

This paper presents a faster sampling method for diffusion based generative models which are usually slow in practice. The key idea is based a progressive distillation approach (e.g., how to distill a 4 step sampler into a 1 step sampler). The paper studies the various design choices for diffusion models which existing work hasn't looked at that deeply and sheds light on the effects of these choices. The paper also shows that DDIM can be seen as a numerical integrator for probability flow ODE. The experimental results are impressive.

There were some concerns such as the effect of progressive distillation and the overhead of distilling the diffusion model but the authors provided a satisfactory response and backed it up with additional results.

Overall, this is a nice paper on making diffusion based generative models generate faster samples and also provides novel insights into the behavior of these models under various design choices. Given the significant recent interest in these models which are pretty impressive in terms of generation quality but slow, the paper indeed makes a timely contribution which will fuel further interest in these models.  All the reviewers have voted for acceptance. Based on my own reading, the reviewers' assessments, the discussions, and the authors' response, I would vote for acceptance.